# ON THE DIVERSITY OF SYNTHETIC DATA AND ITS IMPACT ON TRAINING LARGE LANGUAGE MODELS

## ABSTRACT

The rise of Large Language Models (LLMs) has accentuated the need for diverse, high-quality pre-training data. Synthetic data emerges as a viable solution to the challenges of data scarcity and inaccessibility. While previous literature has focused predominantly on the quality and quantity of real data, our work enables the measurement of diversity in synthetic data and explores its impact on LLM performance. We study the downstream effects of synthetic data diversity during both the pre-training and fine-tuning stages by introducing a new diversity metric, *LLM cluster-agent*, designed to evaluate the diversity of synthetic datasets. Through a series of controlled experiments with models of 350M and 1.4B parameters, we demonstrate that the proposed cluster-based LLM scoring of diversity correlates positively with both pre-training and supervised fine-tuning performance. Our findings also reveal that synthetic data diversity in pre-training affects supervised fine-tuning more significantly than pre-training itself, even for smaller models. We hope this study advances our understanding of the optimal use of synthetic data in LLM training and opens new avenues for efficient data generation processes.

## 1 INTRODUCTION

A common hypothesis behind the success of Large Language Models (LLMs) (Radford et al., 2019; Brown et al., 2020; OpenAI, 2023a;b; Touvron et al., 2023b) is the scaling law of compute, model size, and, most importantly, high-quality pre-training data (Kaplan et al., 2020a; Wei et al., 2022; Muennighoff et al., 2024). State-of-the-art LLMs are often trained on trillions of tokens (Bai et al., 2023; Dubey et al., 2024; OpenAI, 2023b), yet acquiring such massive high-quality corpora has become increasingly difficult (Villalobos et al., 2022). As a remedy, synthetic data have been widely adopted, offering scalability and controllable quality (Bauer et al., 2024; Liu et al., 2024b; Long et al., 2024a). For example, the Phi series (Gunasekar et al., 2023a; Li et al., 2023; Javaheripi et al., 2023; Abdin et al., 2024) leveraged large-scale textbook-style synthetic data combined with real data to achieve strong performance with smaller models. Similarly, synthetic data in programming and math have been used to enhance coding and reasoning abilities (Guo et al., 2024; Yu et al., 2023; Shao et al., 2024), while other works explored synthetic data for supervised fine-tuning (Zelikman et al., 2022; Huang et al., 2022; Liu et al., 2023; Eldan & Li, 2023; Chen et al., 2024b; Huang et al., 2024), instruction tuning (Wang et al., 2022; Xu et al., 2023; Li et al., 2024c; Wang et al., 2024; Chan et al., 2024; Li et al., 2024a;b; Wu et al., 2024), downstream transfer (Meng et al., 2022; Ye et al., 2022), and evaluation (Zhu et al., 2023; 2024a;b).

Despite this wide adoption, understanding *what aspects of synthetic data influence LLM performance and how* remains largely unexplored, particularly for pre-training. Prior studies on real data show that both quality and quantity matter (Kaplan et al., 2020a; Sorscher et al., 2022), with quantity effects repeatedly validated at scale (Radford et al., 2019; Brown et al., 2020; Computer, 2023; Touvron et al., 2023b; Dubey et al., 2024). In contrast, quality factors—including corruption (Elazar et al., 2023), bias (Gallegos et al., 2024), toxicity (Bender et al., 2021), duplication (Lee et al., 2021; Xue et al., 2024), and diversity (Tirumala et al., 2023b)—are harder to isolate due to their interdependence (Kreutzer et al., 2022; Longpre et al., 2023b). Recent work emphasizes that quality outweighs quantity for real data (Soldaini et al., 2024; Penedo et al., 2023; Groeneveld et al., 2024; Tan & Wang, 2024a; Deitke et al., 2024), yet it remains unclear whether these conclusions extend to synthetic data pre-training.

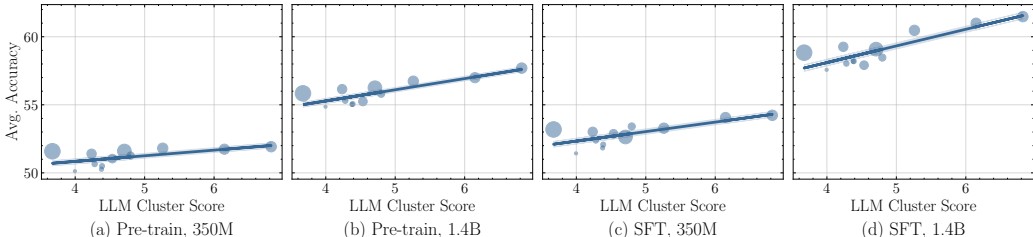

Figure 1: Linear regression of LLM cluster score and benchmark performance of (a) pre-trained 350M; (b) pre-trained 1.4B; (c) supervised fine-tuned 350M; and (d) supervised fine-tuned 1.4B models. Each scatter represents a synthetic dataset with size corresponding to the number of tokens.

In this paper, we propose to study the diversity, as one of the most important quality factors (Tirumala et al., 2023b; Sachdeva et al., 2024), of the pre-training synthetic data. Existing studies on synthetic data in pre-training either only present methods of creating them (Allal et al., 2024b;a) or provide findings that are restricted to relatively small scales (Wu et al., 2022; Allen-Zhu & Li, 2023b; Ye et al., 2024; Zhu & Li, 2023; Allen-Zhu & Li, 2023a; Yang et al., 2024b), with limited understanding on how exactly diversity of the synthetic tokens affect the training of LLMs. However, studying the diversity of synthetic data presents two main challenges. First, the lack of an effective metric for measuring the diversity of text data (Lee et al., 2023; Shaib et al., 2024a; Tirumala et al., 2023a; Ankner et al., 2024), and second, the difficulty of conducting controlled large-scale experiments with synthetic tokens due to the high cost of generation and various aspects influencing their diversity.

To address the limitations of existing diversity metrics, we propose *LLM Cluster-agent*, a pipeline that automatically directs LLMs to cluster a text corpus. The pipeline uses prompts that guide LLMs to summarize key characteristics from randomly sampled data points that capture corpus-level diversity, followed by clustering with a self-verification mechanism. From these results, we compute an *LLM cluster score* as a diversity measure. Packaged as a toolkit, LLM Cluster-agent provides a scalable, consistent, and effective alternative where traditional metrics often fail and yield inconsistent results. For controlled experiments, we extract 620,000 topics from Wikipedia to seed synthetic generation, creating datasets with varied diversity induced by distributions, prompts, models, and real-to-synthetic token ratios. As the first large-scale study on synthetic data diversity, we pre-train language models with 350M and 1.4B parameters on a mix of 34B real and synthetic tokens and conduct supervised fine-tuning to assess downstream effects.

We show that:

- The LLM cluster score correlates positively with both pre-training and fine-tuning performance (Fig. 1), indicating its potential as a practical predictor for large-scale synthetic data pre-training.
- The distribution of synthetic data matters: more unique topics improve diversity, while excessive generations per topic introduce redundancy and hurt performance (Section 3.3).
- Prompts incorporating varied styles and audiences significantly boost diversity and model performance, outperforming Cosmopedia v0.1 and v0.2 baselines (Section 3.4; Allal et al., 2024b;a).
- Stronger LLMs generate more diverse and effective synthetic data; e.g., GPT-4o outperforms GPT-3.5, and Llama-3.1-8B surpasses Mistral-7B (Section 3.5).
- A balanced ratio of real to synthetic tokens yields the best results, while over-weighted synthetic tokens degrade performance due to reduced diversity (Section 3.6).
- Larger diversity continues to benefit fine-tuning even when smaller models saturate in pre-training performance (Fig. 1, Section 3.7).

These findings highlight the practical value of our diversity metric and provide guidance for generating more effective synthetic data for LLM training.

## 2    METRICS FOR MEASURING SYNTHETIC DATA DIVERSITY

Measuring the diversity in large-scale text data is very challenging due to the complex nature of language (Lee et al., 2023; Shaib et al., 2024a). Different metrics have previously been used to measure the diversity of text data, and we broadly categorize them into two types: *heuristic-based* and *model-based*. Heuristic-based metrics, such as vocabulary size, n-gram diversity (Li et al., 2022a; Meister et al., 2023), and self-repetition score (Salkar et al., 2022), often provide a very

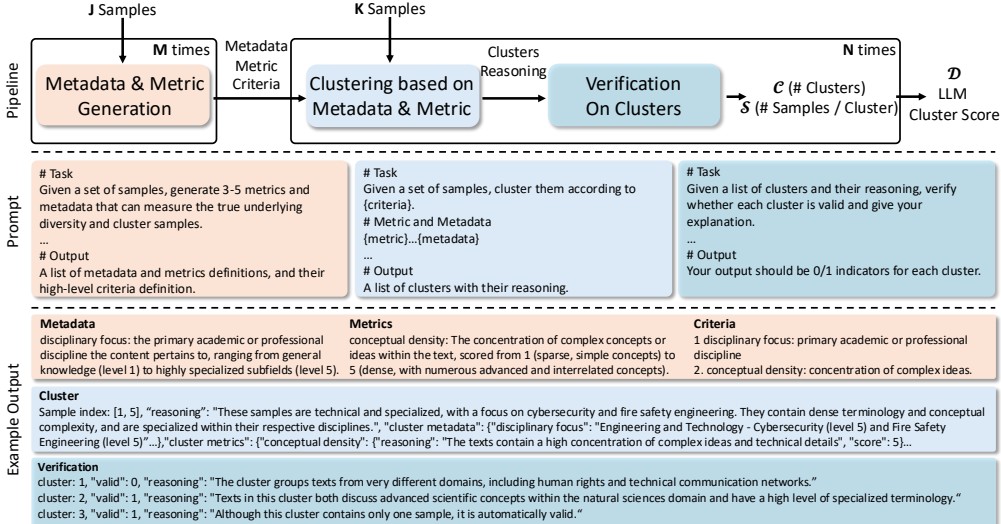

Figure 2: Pipeline, prompt, and example outputs of the proposed LLM Cluster-agent. LLM Cluster-agent first generates metadata and metrics with attributes and scores that captures the underlying distribution and then uses these criteria to perform clustering with an extra self-verification step.

limited view, focusing only on statistical variations within the text without capturing deeper semantic nuances. Model-based methods such as K-means clustering (Abbas et al., 2023) and homogenization score (Lin & Och, 2004; Shaib et al., 2024a) struggle with large-scale and context-rich datasets, as they rely on predefined features, which can oversimplify the true diversity present in the data. These limitations are further compounded in synthetic text data generated by LLMs due to similar patterns in part-of-speech tagging and syntax often present in them (Rosenfeld & Lazebnik, 2024; Shaib et al., 2024c), making it difficult to assess diversity accurately. This motivates us to address the gap by proposing an LLM-based metric to uncover the intricate and latent structures within the data.

## 2.1 LLM CLUSTER-AGENT

Given a text corpus $X = \{x_i\}$ with in total $|X|$ text samples, to allow LLMs to measure their diversity, we propose to originate the measure from the principle of entropy, i.e., capture the underlying distribution of clusters and cluster sizes. However, there are two challenges that prevent LLMs from performing clustering directly. First, it is difficult to define the proper criteria for LLMs to cluster that captures the true distribution. Second, due to the limited context length of LLMs[1], one cannot directly feed the entire text corpus to LLMs for clustering as in traditional clustering methods.

We thus introduce *LLM Cluster-agent*, a diversity measure pipeline that leverages LLM's abilities to interpret semantic meanings and to understand rich contexts of text samples for clustering. To overcome the above challenges, we design LLM Cluster-agent to perform an iterative clustering based on $K$ text samples each time, according to the clustering criteria that are also summarized by the LLM. More specifically, our method includes the following steps, as shown in Fig. 2.

**Metadata and metric generation**. We design two clustering criteria: *metadata* and *metrics*. Metadata guide LLMs to summarize text attributes, while metrics provide scoring rules and reasoning. For efficiency, we sample $J$ corpus examples and prompt the LLM to generate 3–5 metadata and metrics, repeating this $M$ times. A gathering prompt then aggregates the most frequent outputs, which define the clustering criteria. To emphasize focus in the next step, a criteria summary prompt highlights the high-level definitions at the top of the clustering prompt.

**Cluster generation and verification**. Given the criteria, we randomly select $K$ samples and prompt the LLM to form clusters, producing counts $\mathcal{C}_i$ and sample sizes $\mathcal{S}_i$ over $N$ generations. Each cluster includes reasoning, which is checked by a verification prompt to filter invalid results. We define the *LLM Cluster score* as $\mathcal{D} = \frac{1}{N} \sum_{i=1}^{N} \frac{\mathcal{C}_i}{\mathcal{S}_i}$, where $\mathcal{D}$ measures diversity by averaging the ratio of

---

[1]Although LLMs nowadays can support 128K context length or even more, the quality of response usually degenerates as the context length increases (Liu et al., 2024a).

Table 1: Summary of existing and ours diversity metrics.

| Metric | Formulation | Type | Reference |
|---|---|---|---|
| Context Length | $\frac{1}{N}\sum_{i=1}^{N}\|x_i\|$ | Heuristic | - |
| Self-Repet. | $\log\left(k\sum_{i=1}^{k}(\hat{N}_i + 1)\right)$ | Heuristic | Salkar et al. (2022) |
| N-gram Div. | $\frac{\text{Unique n-grams in } X}{\text{Total n-grams in } X}$ | Heuristic | Padmakumar & He (2023); Adelani et al. (2021); Li et al. (2022a) |
| Comp. Ratio | $\frac{\text{Orig. size of } X}{\text{Comp. size of } X}$ | Heuristic | Shaib et al. (2024b) |
| Perplexity | $2^{-\frac{1}{\|X\|}\sum_{i=1}^{\|X\|}\log_2 P_{\text{GPT-2-L}}(x_i)}$ | Model | Ankner et al. (2024) |
| Perplexity Gap | $\|\text{PPL}_{\text{GPT-2-L}} - \text{PPL}_{\text{GPT-2-XL}}\|$ | Model | - |
| K-means | Train.: $\min_{\mu_i}\sum_i^k\sum_{x_j}\|x_j - \mu_i\|^2$ 
 Infer.: $i = \arg\min_i \|x_j - \mu_i\|^2$ | Model | Abbas et al. (2023); Sachdeva et al. (2024) |
| LLM Cluster | $\mathcal{D} = \frac{1}{N}\sum_{i=1}^{N}\frac{\mathcal{C}_i}{\mathcal{S}_i}$ | Model | - |

clusters to samples across generations. This captures variation in themes, topics, and styles within the synthetic corpus. Full prompts are shown in Appendix D, with ablations in Appendices B.3 and B.4.

## 2.2 BASELINE METRICS

We include several commonly used heuristic-based and model-based diversity metrics as baselines (Shaib et al., 2024a). *Context Length (CL)* measures the average token length of the text corpus. *Self-Repetition Score (SRS)* quantifies the repetition of tokens within sentences, while *N-Gram Diversity Score (NDS)* measures the proportion of unique $n$-grams. *Compression Ratio (CR)* compares the g-zip compressed size of the dataset to its original size. *Perplexity* measures the uncertainty of a pre-trained model in predicting the next token and *Perplexity Gap* calculates the perplexity difference between a larger and a smaller model. *K-means Clustering* utilizes feature embeddings from a pre-trained model to cluster the data. A summary of the diversity metrics is shown in Table 1 and we further describe these diversity metrics in Appendix C. Apart from our baseline measures to quantify the diversity of pre-training data, there are other measures, such as the Homogenization Score (Lin & Och, 2004; Shaib et al., 2024b) based on ROUGE-L (Lin, 2004), BERTScore (Zhang et al., 2019), Hypergeometric Distribution D (McCarthy & Jarvis, 2010), and Part-of-Speech Compression Ratio (POS-CR) (Shaib et al., 2024b). However, these metrics are generally computationally prohibitive. Due to this computational and experimental limitation, we do not include these metrics in our study.

## 3 SYNTHETIC DATA DIVERSITY IN PRE-TRAINING

With the proposed LLM Cluster-agent metric, we conduct a series of controlled experiments by generating synthetic data with various levels of diversity and training models on them. We reveal a linear correlation between the LLM Cluster Score and training performance from the perspectives of underlying distribution, prompts and models for generation, and ratio of real and synthetic tokens.

### 3.1 EXPERIMENTS SETUP

**Pre-training**. We adopt the Llama architecture (Touvron et al., 2023b) with a context length of 2,048 and the Codegen-Mono (Li et al., 2023; Nijkamp et al., 2022) tokenizer with a vocabulary size of 50,304. We primarily use 350M and 1.4B models and pre-train all models on the combination of real and synthetic data, except for the baselines on real data only. For real data, we use filtered web data, consisting of the Wikipedia subset and part of the C4 (Raffel et al., 2019) subset of Dolma (Soldaini et al., 2024), code data, consisting of the filtered the Stack (Kocetkov et al., 2022), StackOverflow, and Code Contest (Li et al., 2022b) as in Phi-1.5 (Li et al., 2023), and math data from the filtered OpenWebMath (Paster et al., 2023) subset of Dolma. The real data in total contain 34B tokens, where the ratio of web, code, and math tokens is 4:1:1. For synthetic data, we generate variants with different underlying distributions, prompts, and models for generation (more details in the following sections). Our experiments mainly involve two ratios of real (web) and synthetic tokens: 4:1 for

smaller synthetic data experiments, and 1:1 for larger ones, following Phi-1.5. More ratios are also studied. We train 350M and 1.4B models for a total of 50B and 150B tokens, respectively.

**Supervised Fine-tuning**. In addition to pre-training, we also conduct supervised fine-tuning (SFT) to study the effect of diversity in pre-training data inherited to downstream performance (Chen et al., 2024a). After pre-training the models, we supervised fine-tune them for 3 epochs on the combination of GPT-4 filtered version of the Alpaca (Taori et al., 2023) and FLANv2 (Longpre et al., 2023a). The learning rate of the AdamW optimizer for fine-tuning is set to $2e$-5 and weight decay to 0.

**Benchmark Evaluation**. To evaluate the performance of both the pre-trained model and supervised fine-tuned model, we use WinoGrande (Pîrtoacă et al., 2019), ARC-Easy (Pîrtoacă et al., 2019), ARC-Challenge (Ferré, 2021), BoolQ (Clark et al., 2019), SIQA (Bauer & Bansal, 2021), PiQA (Bisk et al., 2020), HellaSwag (Zellers et al., 2019), and COPA (Roemmele et al., 2011). We report the zero-shot accuracy using LM-Eval Harness (Gao et al., 2021) for both pre-trained and supervised fine-tuned models. We utilized a system prompt consistent to fine-tuning to evaluate tuned models.

**Diversity Evaluation**. To effectively evaluate the diversity of the large-scale synthetic corpus, we employ bootstrapping to obtain robust results. Specifically, we randomly select one million text samples from the corpus and run the baseline diversity metrics and our proposed LLM cluster metric on this subset. We repeat the process for 10 rounds with different random seeds and report the average results and the corresponding error bar. For the model-based metrics, we use BERT-L (Devlin, 2018) embeddings for K-means clustering, and GPT-2-L and GPT-2-XL (Radford et al., 2019) to calculate perplexity and perplexity gap. For K-means clustering, we set the number of clusters to $10K$, which we find as a good trade-off between speed and accurate measurement. We set $K = 10$ and $N = 5K$ for the proposed LLM Cluster-agent. We also find $J = 5$ and $M = 100$ is good enough to obtain meaningful clustering criteria, as we show in Appendix B.4. We use non-uniform scale and mainly compare the relative trend to measure the diversity. More details of the model architecture, training parameters, and evaluation datasets are shown in Appendix A.

## 3.2 Seeding Synthetic Data Generation

To ensure both reasonable quality and diversity of the synthetic data generation, we mainly adopt GPT-4o as the base model for the generation of synthetic text data and utilize a set of pre-defined topics as our generation seeds. The topic generation seeds are obtained by first scrawling the web pages from Wikipedia and then prompting GPT-4 to extract a hierarchy of topics and a set of keywords covered in the content of the page. A visualization of the most frequent topics (and their sub-topics) is shown in Fig. 3. We further run a de-duplication process on all the topics collected and obtain in total 620,000 topics to ensure the wide coverage of knowledge in synthetic data. More detailed distribution and examples of topic seeds and keywords are shown in Appendix E. Our synthetic data generation is based on these topic seeds and keywords in the following experiments.

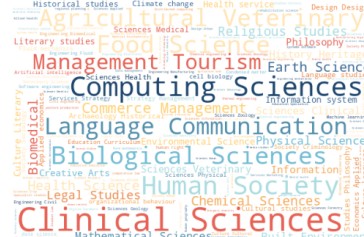

Figure 3: Top topic seeds.

## 3.3 On the Underlying Distribution of Synthetic Data

We first study the effect of the underlying distribution of synthetic data on LLM's performance, i.e., the number of topics $\mathcal{T}$ the and number of generations per topic $\mathcal{G}$ used for synthetic data generation.

**Synthetic Data Generation**. To generate the synthetic data with varying underlying distribution, we sample $\mathcal{T} \sim \{100K, 300K\}$ seeding topics and perform $\mathcal{G} \sim \{10, 20, 30\}$ textbook-style data generation using a simple prompt template that specifies the topic and keywords for each generation. Following the setup of experiments in Phi-series, we also generate a question with answers and step-by-step explanations based on the content at the end of each synthetic sample. We refer to this prompt template as *Topic*. The detailed prompt template and output examples are shown in Appendix F. We

Table 2: Synthetic token counts of varying underlying topics $\mathcal{T}$ and generations $\mathcal{G}$.

| $\mathcal{T}$ | | 100K | | | 300K | |
|---|---|---|---|---|---|---|
| $\mathcal{G}$ | 10 | 20 | 30 | 10 | 20 | 30 |
| # Tokens (B) | 0.58 | 1.01 | 1.48 | 1.74 | 3.04 | 4.43 |

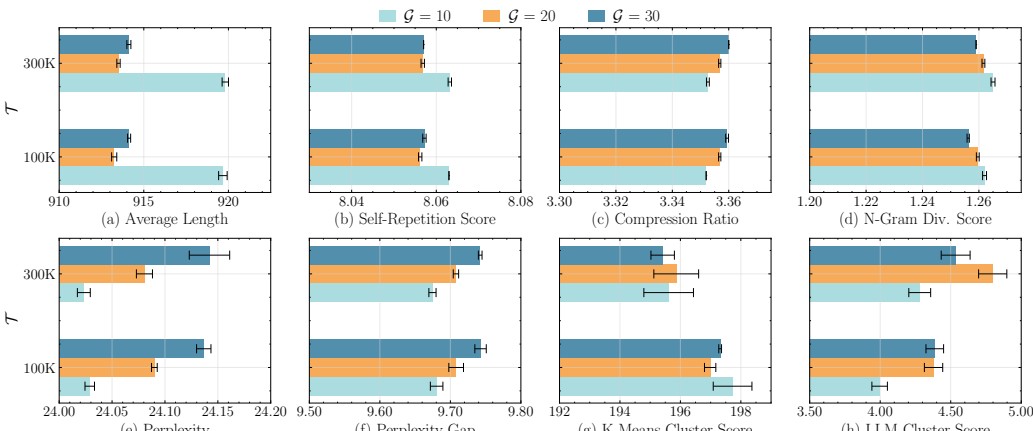

Figure 4: Diversity results of varying underlying number of topics ($\mathcal{T}$) and number of generations per topic ($\mathcal{G}$) in synthetic data. (a) Average length of synthetic samples; (b) Self-repetition score; (c) Compression ratio; (d) N-gram diversity score; (e) Perplexity of GPT-2-L; (f) Perplexity gap between GPT-2-L and GPT-2-XL; (g) K-means cluster score of BERT-L embeddings; (g) LLM cluster score. Ours demonstrates the most significant difference in diversity, aligning with the underlying topic distribution. It also reflects the saturated and deteriorated diversity as $\mathcal{G}$ increases.

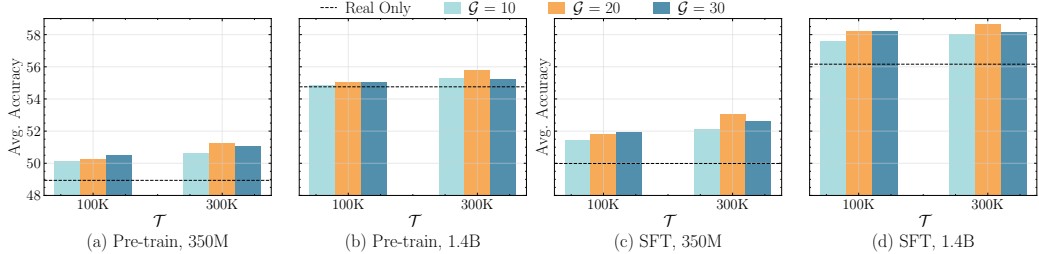

Figure 5: Benchmark average accuracy of pre-trained and supervised fine-tuned 350M and 1.4B models by varying underlying number of topics ($\mathcal{T}$) and number of generations per topic $\mathcal{G}$ in synthetic data. The performance of both pre-trained and supervised fine-tuned models well aligns with our LLM cluster diversity metric: first increases and then saturates or deteriorates with diversity.

present the token count of the synthetic data generated using this prompt in Table 2. For fair comparison, we increase the sampling weight to make the effective synthetic tokens as 4.5B, and combine with the 34B real tokens for pre-training the models.

**Results**. After generating the synthetic data, we perform the diversity evaluation on them and report the results of different diversity metrics in Fig. 4. Although baseline metrics might be able to measure the diversity of different datasets from various domains or model outputs, as reported by Shaib et al. (2024a), they cannot discriminate the underlying distribution of synthetic data well, with trivial differences present in the metric values. Similar observations persist even for model-based metrics such as perplexity and perplexity gap (Ankner et al., 2024). One can also find that the traditional clustering method, i.e., K-means clustering, fails to capture the diversity of the underlying distributions, where the cluster score of synthetic tokens with 300K topics is measured to be smaller than that of 100K topics. More importantly, the diversity measured by both the heuristic-based and model-based baseline metrics demonstrates different trends, which is difficult to interpret.

In contrast, the proposed LLM cluster metric presents a more significant difference in the diversity of synthetic tokens, where the data with 100K topics generally show less diversity compared to that of 300K. LLM cluster score also tends to increase first and then decrease as $\mathcal{G}$ increases, showing saturated or even deteriorated diversity. This has not been observed in any of the baseline diversity metrics. More interestingly, in the average benchmark results of both pre-trained and supervised fine-tuned models, as shown in Fig. 5, the performance highly aligns with our LLM cluster diversity measure. Our results suggest that diversity, in terms of the number of topics $\mathcal{T}$ and the proper number of generations per topic $\mathcal{G}$, in synthetic data pre-training is essential for better performance.

## 3.4 PROMPTS FOR SYNTHETIC DATA GENERATION

In this part, we continue our study with different prompt templates for generating more diverse synthetic data. As suggested in the creation of Cosmopedia-v0.1 (Allal et al., 2024b) and Cosmopedia-v0.2 (Allal et al., 2024a), the prompt template used for the generation of synthetic tokens is also very important for performance. However, it is unclear on what dimension the diversity of synthetic data can better increase, and we try to conclude an answer from a set of controlled experiments.

**Synthetic Data Generation**. To design prompts from different diversity dimensions, we start from the *Topic* prompt template used in Section 3.3. We first increase the dimension of styles of the

Table 3: Synthetic token counts of varying generation prompts.

| Prompt | Cosmopedia v0.1 | Cosmopedia v0.2 | Topic | Topic Styles | Topic Styles Persona | Multi-Topic Styles Persona |
|---|---|---|---|---|---|---|
| # Tokens (B) | 22.09 | 28.60 | 10.44 | 12.64 | 12.90 | 12.27 |

synthetic text, including textbook narrative, textbook academic, blogpost, and wikihow, similar to Cosmopedia v0.1. We term this prompt template as *Topic Style*. Based on it, we further expand the targeted audience of the synthetic content. In contrast to Cosmopedia, which adopted a limited number of audiences, we utilize the recent advance of personas for the creation of synthetic content (Chan et al., 2024). For each generation, we randomly sample a set of personas and let GPT-4o to select the most appropriate one as the target audience for the generation. This prompt is thus referred to as *Topic Styles Persona*. Lastly, we further introduce multiple topic seeds in the prompt template, instead of just a single topic, and let GPT-4o select a combination of topics for content creation. We term this prompt as *Multi-Topic Styles Persona*. We use these four prompt variants to generate around 10-12B synthetic tokens utilizing the underlying 620K topics, and pre-train models by up-weighting the synthetic tokens as in total 20B, similarly to Phi-series. In addition, we also pre-train models on Cosmopedia v0.1 and Cosmopedia v0.2 as our large-scale synthetic data baselines, which are down-weighted to 20B for fair comparison. The token statistics are shown in Table 3, and the details, examples, and outputs of the prompt template variants are shown in Appendix F.

**Results**. We present diversity measurements of synthetic data from different prompt templates in Fig. 6. Baseline heuristic and model-based metrics show inconsistent diversity across datasets. Benchmark results for the 350M and 1.4B models are given in Fig. 7, where both pre-trained and supervised fine-tuned models correlate well with the LLM cluster score. Notably, although Cosmopedia v0.2 was generated with better-optimized prompts (Allal et al., 2024a), it shows lower diversity than v0.1, leading to weaker model performance. Our *Topic* prompt template matches v0.1 with over 50% fewer tokens. Other variants yield both higher diversity and superior performance than Cosmopedia. Interestingly, the *Multi-Topic Styles Persona* template reduces diversity compared to *Topic Styles Persona*, likely due to redundancy from combining multiple topics. Overall, our results suggest that incorporating personas (Chan et al., 2024) in synthetic data generation enhances diversity and, in turn, model performance.

## 3.5 MODELS FOR SYNTHETIC DATA GENERATION

**Synthetic Data Generation**. We study the diversity of synthetic tokens generated by different models in this part. In previous sections, we default our synthetic generation model as GPT-4o. Here, we compare the synthetic generation using GPT-3.5, and two open-source models: Llama-3.1-8B-Instruct (Dubey et al., 2024) and Mistral-7B-Instruct[2] (Jiang et al., 2023). From our previous results, we use the same *Topic Styles Persona* prompt

Table 4: Synthetic token counts of models.

| Model | GPT-4o | GPT-3.5 | Llama-3.1 | Mistral |
|---|---|---|---|---|
| # Tokens (B) | 5.00 | 4.62 | 4.04 | 4.39 |

template for the synthetic generation with different models. Similarly to Section 3.3, we up-weight the generated synthetic tokens to 5B for pre-training, whose statistics are shown in Table 4. We select 5B tokens from our corresponding GPT-4o generation in Section 3.4 as an additional comparison. We also set an additional variant with mixed synthetic data from all models. The output examples are shown in Appendix F. Here, we only pre-train and supervised fine-tune 350M models and report the LLM cluster score measurement mainly due to the computational limits.

---

[2]While Cosmopedia (Allal et al., 2024b;a) mainly used Mistral-8x7B-Instruct for synthetic data generation, we instead select smaller models here mainly due to the computational limit.

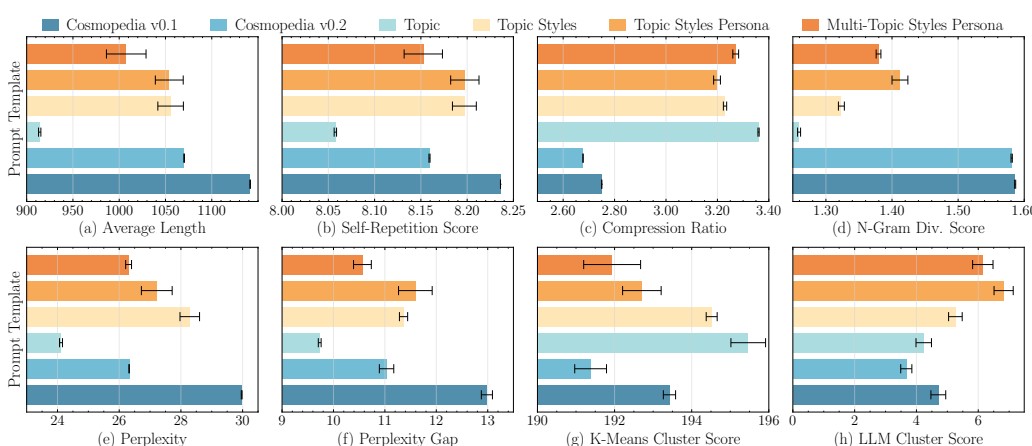

Figure 6: Diversity results of synthetic data generated by various prompt templates. (a) Average length of synthetic samples; (b) Self-repetition score; (c) Compression ratio; (d) N-gram diversity score; (e) Perplexity of GPT-2-L; (f) Perplexity gap between GPT-2-L and GPT-2-XL; (g) K-means cluster score of BERT-L embeddings; (g) LLM cluster score. The baseline metrics show inconsistent measures of diversity, whereas the proposed LLM cluster method well captures the diversity.

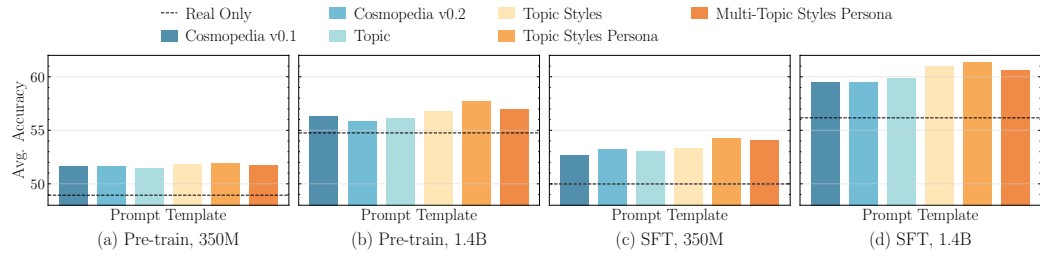

Figure 7: Benchmark results of pre-trained and supervised fine-tuned models by varying the prompt templates for synthetic data generation. *Persona* and *Styles* improves diversity and performance.

**Results**. We present both the results of the LLM cluster diversity and the model performance in Fig. 8. One can observe that the synthetic data generated by more capable models usually present better diversity, i.e., GPT-4o over GPT-3.5 and Llama-3.1 over Mistral. This trend is also reflected in the performance of both the pre-trained and supervised fine-tuned models. Mixing up the synthetic data generated by different base LLMs can also slightly improve diversity, leading to better performance. Our results suggest that the use of synthetic data from more advanced models and mixed models can be potentially beneficial in practice.

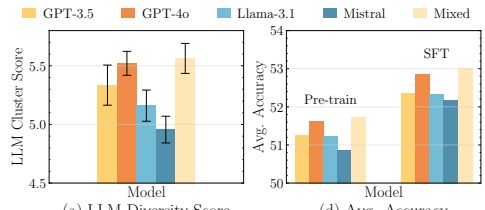

Figure 8: (a) LLM diversity score of synthetic data from different models. (b) Average performance of trained models.

### 3.6 RATIO BETWEEN REAL AND SYNTHETIC TOKENS

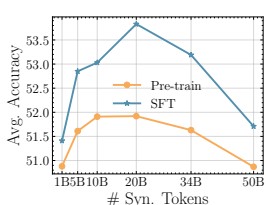

Figure 9: Results of varying real-syn ratio.

Here, we study the effect of the ratio between real and generated synthetic tokens. We re-use the 12.9B synthetic data created by *Topic Styles Persona* prompt template. We train 350M models by adjusting the sampling weight during training to make them effectively 1B, 5B, 10B, 20B, 34B, and 50B. The results are shown in Fig. 9. As we can observe, the accuracy generally improves as the proportion of synthetic tokens initially increases, i.e., from 1B to 20B. However, when the ratio becomes skewed heavily toward synthetic tokens, i.e., over 34B, the average accuracy drops significantly, suggesting that the over-weighting of the synthetic data may introduce redundancy and thus hurt model performance.

### 3.7 DIVERSITY, TOKEN SIZE, AND MODEL SIZE

**Correlations between LLM Cluster Score and Model Performance**. We plot the linear regression of the LLM cluster score and model performance in Fig. 1, demonstrating a positive correlation between them. As the LLM cluster score increases, indicating greater diversity in synthetic data, the average accuracy also improves consistently. This trend is observed for both smaller models (350M) and larger models (1.4B), although the latter generally correlates more with the proposed LLM cluster score as shown in Appendix B.2, suggesting that more capable models benefit more from increased synthetic data diversity.

**Larger Model Requires Larger Diversity**. One can also find that the 1.4B parameter models require and benefit from a higher level of diversity to fully leverage their capacity. As the LLM cluster score increases, larger models show a more pronounced improvement in performance compared to smaller models. Interestingly, while the pre-training performance of smaller models tends to saturate with larger diversity, the supervised fine-tuning performance can still benefit significantly.

## 4 RELATED WORK

Scaling language models in size and data has led to increasingly powerful systems (Kaplan et al., 2020b; Touvron et al., 2023a;b; Jiang et al., 2023; Bai et al., 2023; Yang et al., 2024a; AI et al., 2024; Team et al., 2024). However, high-quality pre-training data are finite and may soon be exhausted (Villalobos et al., 2022). To address this, synthetic data from advanced LLMs are now widely used for pre-training (Gunasekar et al., 2023b; Ben Allal et al., 2024; Allal et al., 2024b; Long et al., 2024b), fine-tuning, and alignment (Wang et al., 2023; Taori et al., 2023; Wu et al., 2024). Beyond scale, the quality of pre-training data plays a critical role in model performance (Sachdeva et al., 2024; Penedo et al., 2024), with diversity shown to be especially important for strong downstream generalization (Miranda et al., 2024; Tirumala et al., 2023a; Chung et al., 2023). Consequently, measuring and ensuring data quality has become central, as noisy or low-quality data can degrade model performance (Penedo et al., 2024).

A range of strategies has been proposed to curate high-quality subsets from large corpora. Simple filtering using LLMs can reduce data size dramatically while maintaining performance (Sachdeva et al., 2024), and perplexity-based selection or diversity-aware sampling has proven effective (Ankner et al., 2024; Tirumala et al., 2023b; Tan & Wang, 2024b; Longpre et al., 2023b). Recent work has also emphasized evaluating quality with metrics such as perplexity, factuality, and alignment with human judgment (Shaib et al., 2024b; Montahaei et al., 2019; Li et al., 2020). Among these factors, diversity remains a key determinant of high-quality pre-training data (Tirumala et al., 2023b), though existing diversity metrics are designed for natural data and face limitations when applied to synthetic corpora.

## 5 CONCLUSION

In this study, we investigated the impact of synthetic data diversity on the performance of LLMs. We proposed and validated a new metric, LLM Cluster-agent, to quantify the diversity of synthetic data. Our experiments demonstrated that increased diversity correlates positively with model performance, particularly in downstream fine-tuning tasks. Moreover, the choice of generation seeds, the prompt template, the generation model, and the ratio between real and synthetic tokens all significantly influence both the data diversity and model performance. Although the scale of models in this study is mainly restricted up to 1.4B due to computational limits, we demonstrated that the results in this study may present the potential to be applied on a larger scale. These results suggest that diverse, high-quality synthetic data is essential for the training of robust and effective LLMs, paving the way for future improvements in the generation and utilization of synthetic data.

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

# Appendix

CONTENTS

## A  TRAINING SETUP

In this section, we provide more details on our training setup.

### A.1  PRE-TRAINING SETUP

For pre-training, we use AdamW optimizer with a linear-warmup-linear-decay learning rate schedule to pre-train the 350M and 1.4B models. The maximum learning rate is set to $3e$-4, betas of AdamW optimizer are set to 0.9 and 0.95, and the weight decay is set at 0.1. We adopt a global batch size of 256 and 128 for 350M and 1.4B models respectively. The 350M models are trained with 16 A100 and the 1.4B models are trained with 32 A100. The 350M models are trained for in total 50B tokens, and 1.4B models are trained for 150B tokens. We use fp16 and Zero-2 of DeepSpeed (Rasley et al., 2020) to speed up training. The model configurations are shown in Table 5.

Table 5: Configuration of 350M and 1.4B models.

| Model Size | Vocab Size | Context Length | Hidden Size | Intermediate Size | # Layers | # Heads | Attn. Dropout |
|---|---|---|---|---|---|---|---|
| 350M | 50340 | 2048 | 960 | 2560 | 28 | 15 | 0.1 |
| 1.4B | 50340 | 2048 | 2048 | 8192 | 16 | 32 | 0.1 |

## B  EXPERIMENTS RESULTS

In this section, we present the detailed benchmark results.

### B.1  MAIN RESULTS

The main experiments results are shown here. We present the details results of Section 3.3 in Table 6, the detailed results of Section 3.4 in Table 7, the detailed results of Section 3.5 in Table 8. For ARC-challenge and HellaSwag, we report 'acc_norm' from LM-Eval-Harness, and 'acc' for other evaluated tasks.

Table 6: Benchmark results of varying underlying distribution.

| Model | $\mathcal{T}$ | $\mathcal{G}$ | Average | Common Sense | | | | | Language Understanding | | |
|---|---|---|---|---|---|---|---|---|---|---|---|
| | | | | ARC-C | ARC-E | BoolQ | SiQA | WinoGrande | PIQA | COPA | HellaSwag |
| 350M | 100 | 10 | 50.12 | 25.85 | 52.69 | 58.04 | 38.28 | 50.75 | 68.34 | 67.00 | 40.02 |
| | | 20 | 50.26 | 25.91 | 52.02 | 56.47 | 38.64 | 52.09 | 67.92 | 69.00 | 40.06 |
| | | 30 | 50.50 | 26.54 | 52.99 | 56.73 | 38.84 | 53.12 | 68.01 | 68.00 | 39.78 |
| | 300 | 10 | 50.65 | 27.30 | 51.85 | 58.93 | 38.54 | 51.30 | 68.44 | 69.00 | 39.85 |
| | | 20 | 51.28 | 27.30 | 51.85 | 58.93 | 39.54 | 52.30 | 68.44 | 72.00 | 39.85 |
| | | 30 | 51.05 | 26.54 | 52.86 | 59.57 | 39.43 | 53.17 | 67.68 | 69.00 | 40.12 |
| 350M SFT | 100 | 10 | 51.43 | 28.33 | 53.93 | 59.78 | 39.10 | 52.09 | 69.81 | 67.00 | 41.41 |
| | | 20 | 51.83 | 28.88 | 53.91 | 60.55 | 39.51 | 52.01 | 70.00 | 68.00 | 41.80 |
| | | 30 | 51.96 | 28.67 | 54.18 | 60.44 | 40.69 | 52.38 | 69.46 | 68.00 | 41.83 |
| | 300 | 10 | 52.38 | 29.16 | 54.28 | 60.04 | 39.30 | 51.85 | 69.23 | 71.00 | 42.19 |
| | | 20 | 53.04 | 29.65 | 54.65 | 60.55 | 39.95 | 52.41 | 70.25 | 74.00 | 42.82 |
| | | 30 | 52.62 | 29.07 | 54.77 | 60.09 | 39.76 | 53.72 | 69.27 | 72.00 | 42.29 |
| 1B | 100 | 10 | 54.86 | 28.24 | 62.29 | 57.41 | 41.74 | 58.88 | 73.67 | 73.00 | 43.66 |
| | | 20 | 55.02 | 28.75 | 62.79 | 59.63 | 42.15 | 57.59 | 73.18 | 72.00 | 44.09 |
| | | 30 | 55.06 | 28.90 | 61.57 | 59.98 | 42.81 | 57.62 | 74.05 | 72.00 | 43.56 |
| | 300 | 10 | 55.30 | 29.52 | 62.12 | 58.54 | 40.70 | 56.27 | 73.29 | 78.00 | 43.95 |
| | | 20 | 55.81 | 30.20 | 63.22 | 59.79 | 41.94 | 59.59 | 73.83 | 73.00 | 44.91 |
| | | 30 | 55.24 | 29.75 | 62.35 | 58.87 | 41.30 | 58.41 | 74.43 | 72.00 | 44.84 |
| 1B SFT | 100 | 10 | 57.57 | 31.63 | 63.68 | 58.56 | 42.10 | 59.38 | 74.14 | 73.00 | 58.08 |
| | | 20 | 58.19 | 31.31 | 64.09 | 58.87 | 42.50 | 59.33 | 74.65 | 76.00 | 58.76 |
| | | 30 | 58.20 | 32.25 | 63.90 | 59.04 | 42.40 | 59.75 | 74.93 | 75.00 | 58.33 |
| | 300 | 10 | 58.03 | 32.57 | 64.31 | 59.99 | 41.15 | 59.35 | 73.89 | 75.00 | 58.01 |
| | | 20 | 58.65 | 34.00 | 65.32 | 60.75 | 42.48 | 59.20 | 74.73 | 74.00 | 58.68 |
| | | 30 | 58.16 | 33.62 | 64.95 | 60.81 | 41.04 | 59.01 | 74.09 | 73.00 | 58.76 |

Table 7: Benchmark results of varying prompt templates.

| Model | Data | Average | Common Sense | | | | | Language Understanding | | |
|---|---|---|---|---|---|---|---|---|---|---|
| | | | ARC-C | ARC-E | BoolQ | SiQA | WinoGrande | PIQA | COPA | HellaSwag |
| 350M | Real Only | 48.94 | 24.40 | 48.78 | 58.96 | 38.59 | 52.09 | 66.81 | 66.00 | 35.88 |
| | Cosmepedia v0.1 | 51.61 | 27.68 | 53.90 | 59.98 | 39.10 | 53.12 | 69.57 | 68.00 | 41.49 |
| | Cosmepedia v0.2 | 51.59 | 28.69 | 54.98 | 59.46 | 38.12 | 51.80 | 68.75 | 70.00 | 40.89 |
| | *Topic* | 51.40 | 28.05 | 54.29 | 60.20 | 38.41 | 53.51 | 67.85 | 68.00 | 40.92 |
| | *Topic Styles* | 51.81 | 28.41 | 56.02 | 60.04 | 39.25 | 53.41 | 68.17 | 68.00 | 41.17 |
| | *Topic Styles Persona* | 51.92 | 28.90 | 55.60 | 60.36 | 39.38 | 53.54 | 69.36 | 67.00 | 41.24 |
| | *Multi-Topic Styles Persona* | 51.74 | 27.90 | 53.87 | 60.17 | 39.46 | 53.04 | 68.87 | 70.00 | 40.59 |
| 350M SFT | Real Only | 50.00 | 27.05 | 52.86 | 58.31 | 39.20 | 51.46 | 66.00 | 67.00 | 38.10 |
| | Cosmepedia v0.1 | 52.64 | 29.56 | 55.80 | 60.28 | 40.97 | 51.80 | 70.57 | 69.00 | 43.41 |
| | Cosmepedia v0.2 | 53.29 | 30.78 | 55.23 | 60.26 | 41.66 | 53.35 | 69.75 | 71.00 | 44.28 |
| | *Topic* | 53.03 | 29.33 | 55.98 | 60.34 | 40.23 | 52.96 | 70.85 | 70.00 | 44.58 |
| | *Topic Styles* | 53.37 | 30.12 | 56.03 | 60.74 | 40.51 | 53.07 | 71.17 | 70.00 | 45.32 |
| | *Topic Styles Persona* | 54.29 | 31.82 | 56.84 | 60.86 | 41.15 | 53.70 | 71.36 | 72.00 | 46.60 |
| | *Multi-Topic Styles Persona* | 54.06 | 31.82 | 56.98 | 60.07 | 41.49 | 52.22 | 70.87 | 73.00 | 46.00 |
| 1B | Real Only | 54.76 | 28.07 | 62.08 | 57.98 | 42.58 | 58.80 | 73.45 | 71.00 | 44.08 |
| | Cosmepedia v0.1 | 56.25 | 29.78 | 64.84 | 58.75 | 42.99 | 59.35 | 73.61 | 75.00 | 45.71 |
| | Cosmepedia v0.2 | 55.84 | 32.08 | 66.37 | 54.81 | 43.60 | 59.04 | 73.67 | 71.00 | 46.16 |
| | *Topic* | 56.15 | 30.12 | 66.04 | 60.92 | 42.58 | 58.93 | 73.88 | 71.00 | 45.73 |
| | *Topic Styles* | 56.74 | 31.83 | 66.62 | 59.85 | 43.97 | 58.64 | 73.01 | 74.00 | 45.96 |
| | *Topic Styles Persona* | 57.82 | 32.46 | 67.20 | 62.65 | 44.51 | 59.97 | 73.98 | 74.00 | 47.80 |
| | *Multi-Topic Styles Persona* | 56.99 | 32.44 | 66.81 | 61.42 | 43.41 | 58.74 | 73.49 | 72.00 | 47.64 |
| 1B SFT | Real Only | 57.16 | 31.31 | 58.75 | 58.96 | 43.07 | 59.43 | 74.06 | 73.00 | 58.08 |
| | Cosmepedia v0.1 | 59.46 | 34.79 | 65.42 | 62.13 | 42.12 | 59.51 | 75.47 | 77.00 | 59.25 |
| | Cosmepedia v0.2 | 59.46 | 34.45 | 66.18 | 63.31 | 43.71 | 59.20 | 75.60 | 72.00 | 61.21 |
| | *Topic* | 59.88 | 34.94 | 66.96 | 64.61 | 43.12 | 59.35 | 74.97 | 74.00 | 61.11 |
| | *Topic Styles* | 60.97 | 35.57 | 67.69 | 65.08 | 43.58 | 59.57 | 75.57 | 78.00 | 62.57 |
| | *Topic Styles Persona* | 61.32 | 35.78 | 68.04 | 65.19 | 44.10 | 60.39 | 76.17 | 78.00 | 62.57 |
| | *Multi-Topic Styles Persona* | 60.59 | 34.36 | 67.93 | 64.79 | 43.11 | 60.01 | 75.39 | 76.00 | 63.03 |

Table 8: Benchmark results of varying synthetic data generation models.

| Model | Gen Model | Average | Common Sense | | | | | Language Understanding | | |
|---|---|---|---|---|---|---|---|---|---|---|
| | | | ARC-R | ARC-E | BoolQ | SiQA | WinoGrande | PIQA | COPA | HellaSwag |
| 350M | Llama-3.1-8B-Instruct | 51.22 | 26.37 | 54.54 | 58.17 | 39.10 | 52.88 | 68.39 | 70.00 | 40.34 |
| | Mistral-7B-Instruct | 50.86 | 26.02 | 54.36 | 58.31 | 39.20 | 51.99 | 67.95 | 69.00 | 40.03 |
| | GPT-3.5 | 51.23 | 26.87 | 53.99 | 59.23 | 38.67 | 52.72 | 68.22 | 70.00 | 40.17 |
| | GPT-4o | 51.61 | 27.13 | 54.53 | 59.65 | 38.71 | 52.93 | 68.45 | 71.00 | 40.51 |
| | Mixed | 51.72 | 26.88 | 54.38 | 59.47 | 39.33 | 52.99 | 68.79 | 71.00 | 40.88 |
| 350M SFT | Llama-3.1-8B-Instruct | 52.32 | 29.65 | 55.51 | 60.52 | 39.71 | 52.17 | 68.74 | 70.00 | 42.25 |
| | Mistral-7B-Instruct | 52.17 | 28.79 | 55.60 | 60.43 | 39.61 | 51.62 | 68.32 | 71.00 | 42.00 |
| | GPT-3.5 | 52.36 | 29.13 | 55.84 | 60.19 | 39.88 | 52.09 | 69.89 | 69.00 | 42.83 |
| | GPT-4o | 52.85 | 29.75 | 56.16 | 60.72 | 39.97 | 52.22 | 70.05 | 71.00 | 42.95 |
| | Mixed | 53.02 | 29.47 | 57.05 | 60.40 | 39.15 | 52.63 | 70.81 | 71.00 | 43.62 |

## B.2 CORRELATION OF METRIC VALUES AND PERFORMANCE

Here, we show more qualitative and quantitative results on the comparison of correlation between the metric values and the performance. As shown in Fig. 11 and Table 9, the proposed LLM-cluster metric demonstrates the best correlation between its diversity score and the performance of LLMs, both on pre-training and supervised fine-tuning benchmark.

Table 9: Pearson correlation coefficients (with p-value) of metric values and performance.

| Metric | Pre-training (350M) | Downstream (350M) | Pre-training (1.4B) | Downstream (1.4B) |
|---|---|---|---|---|
| Self-Repetition Score | 0.5583 (0.0422) | 0.6185 (0.0320) | 0.7471 (0.0052) | 0.6523 (0.0147) |
| Compression Ratio | -0.4798 (0.1144) | -0.2751 (0.3868) | -0.2600 (0.4143) | -0.2941 (0.3533) |
| N-gram Diversity | 0.5878 (0.0444) | 0.4289 (0.1640) | 0.4382 (0.1541) | 0.4378 (0.1545) |
| Perplexity | 0.5066 (0.0101) | 0.5095 (0.0905) | 0.6587 (0.0198) | 0.6761 (0.0157) |
| Perplexity Gap | 0.6773 (0.0155) | 0.4799 (0.1142) | 0.6310 (0.0277) | 0.6203 (0.0313) |
| K-means | -0.8487 (0.0004) | -0.8312 (0.0008) | -0.7400 (0.0059) | -0.7321 (0.0067) |
| LLM-Cluster | 0.5930 (0.0421) | 0.7481 (0.0051) | 0.8457 (0.0005) | 0.7384 (0.0061) |

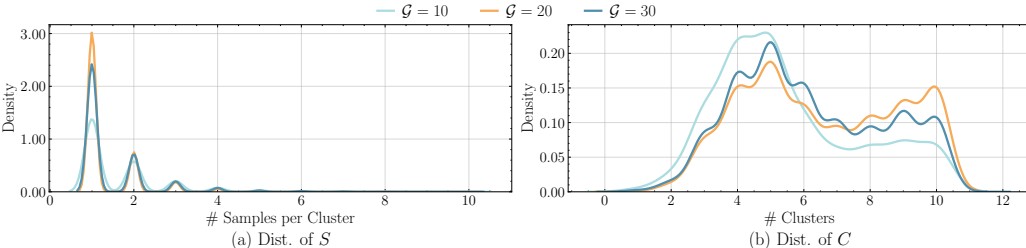

Figure 10: Density estimation of (a) number of samples per cluster $S$ and (b) number of clusters $C$ from LLM cluster results on synthetic data generated with *Topic* prompt using $\mathcal{T} = 300K$, and $\mathcal{G} \sim \{10, 20, 30\}$. LLM Cluster-agent can discriminate the diversity of the underlying distributions.

## B.3 ABLATION STUDY OF LLM CLUSTER METRIC

**Pipeline Parameters**. We conduct ablation experiments on $K$ and $N$, and $J$ and $M$, with ablation results present in Appendix B.4 due to the space limit. We show that the generation of metadata and metric is robust to the parameters $J$ and $M$. The clustering performance decreases with very small and large $K$, and saturates as $N$ increases, showing the scalability of proposed metric.

**Pipeline Components**. We also conduct ablation on the components of the pipeline. We compare the LLM cluster results using the entire pipeline, the pipeline without the verification component, and only the clustering component with manually defined metadata and metrics. The results in Appendix B.4 demonstrate that metadata and metrics generation is essential to guarantee reasonable clustering performance, and the self-validation step can further boost the clustering performance.

**Different LLMs**. We perform an additional ablation on the models used in the proposed LLM clustering pipeline, i.e., GPT-4, GPT-4o, GPT-3.5, and Llama-3.1. From the results, we can observe that different LLMs often present consistent and robust clustering results using the proposed pipeline.

**Distribution of Clusters**. We plot the distribution of $\mathcal{C}$ and $\mathcal{S}$ of our LLM cluster score results in Section 3.3 with $\mathcal{T} = 300K$, as shown in Fig. 10. We can observe that, from the density of $\mathcal{C}$ and $\mathcal{S}$, LLM Cluster-agent can capture the nuanced diversity difference of the underlying distribution.

## B.4 ABLATION RESULTS

Here, we provide all of our ablation results on the proposed LLM Cluster-agent.

The ablation on $J$ and $M$ are shown in Table 10. We show that $J = 5$ and $M = 100$, and with larger values of these two parameters, produce quite consistent top metadata and metrics that will be used for clustering criteria.

Table 10: Ablation of $J$ and $M$ on top-3 metadata and metrics.

| J | M | Top3 Metadata | Top3 Metric |
|---|---|---|---|
| 5 | 10 | Analysis Technique, Industry Relevance | Clarity of Explanation, Jargon Usage, Technicality Level |
| 5 | 50 | Temporal Relevance, Technical Concept Depth, Terminology Density | Semantic Coherence, Technical Language Density, Contextual Depth |
| 5 | 100 | Subject Domain, Conceptual Density, Narrative Structure | Terminology Density, Interdisciplinary Index, Practical Impact Factor |
| 5 | 500 | Disciplinary Focus, Conceptual Density, Interdisciplinary Relevance | Interdisciplinary Integration, Conceptual Density, Lexical Diversity |
| 3 | 100 | Domain Specificity, Conceptual Complexity, Semantic Complexity | Novelty Score, Practical Impact Factor, Conceptual Clarity |
| 10 | 100 | Disciplinary Focus, Conceptual Density, Terminology Density | Interdisciplinary Integration, Information Density, Lexical Diversity |
| 15 | 100 | Disciplinary Focus, Text Complexity, Narrative Style | Interdisciplinary Integration, Conceptual Density, Lexical Diversity |
| 30 | 100 | Discipline Focus, Text Complexity, Textual Cohesion | Interdisciplinary Integration, Novelty Index, Lexical Diversity |
| 50 | 100 | Interdisciplinary Relevance, Domain Specificity, Sample Source Origin | Jargon Richness, Informativeness, Audience Breadth |

The ablation of clustering score results about parameters $K$ and $N$ are shown in Table 11(a) and Table 11(b), pipeline components are shown in Table 11(c), and generation models are shown in Table 11(d). One can observe that $K = 10$ produce the most robust clustering results, where smaller and larger $K$ present larger variations in results. We also show that with sufficient large $N$ as 5K or 10K, the clustering results becomes stable. For the components, we find that both the metadata and metric generation and self-verification step is essential to achieve reasonable clustering performance. We also demonstrate that the proposed metric is robust to the generation models.

We additionally provide an ablation study on the self-verification module. In Table 12, we perform a human evaluation on the invalid filtered clusters from the self-verification module, and find that a

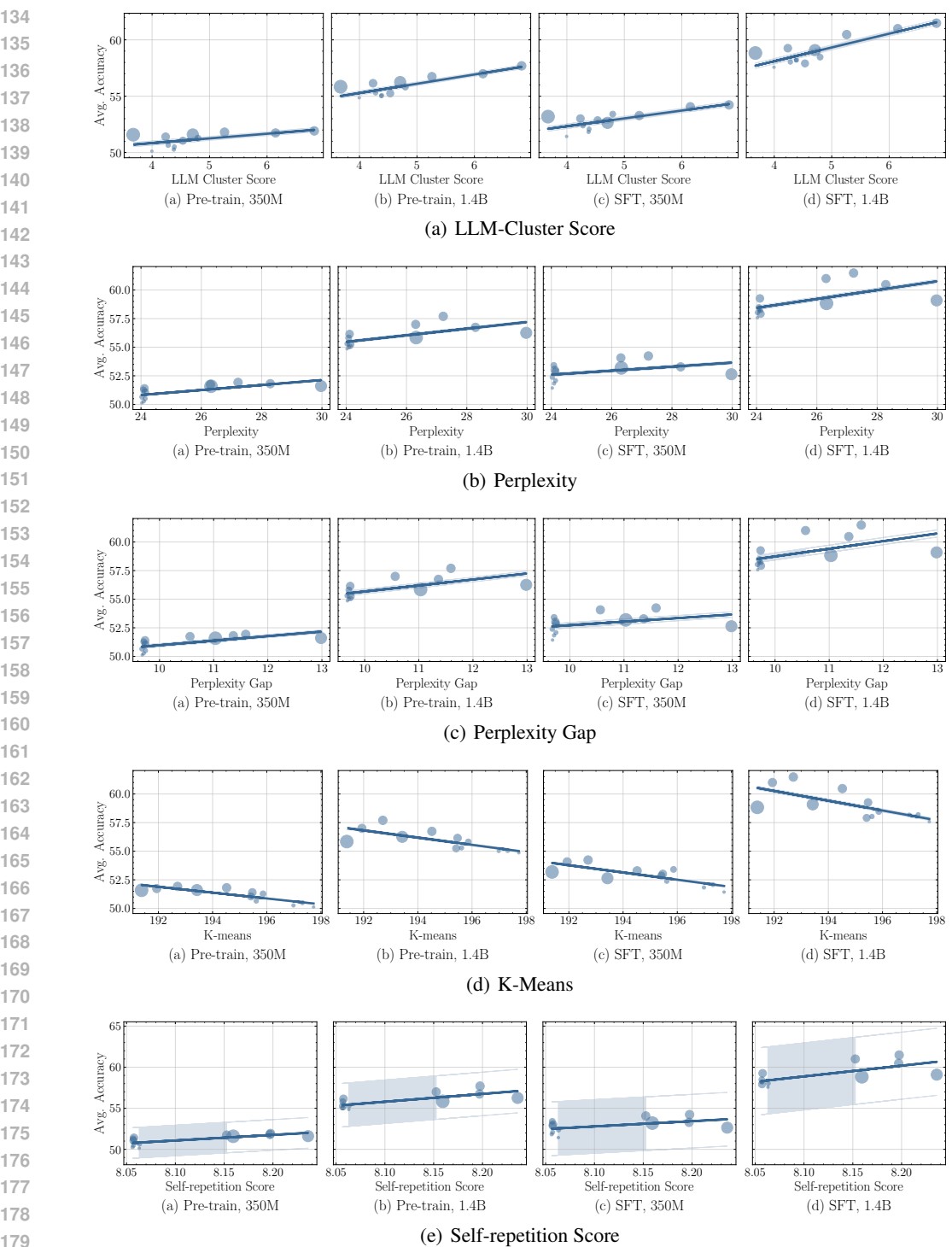

Figure 11: Correlation between the metric values and the performance

large proportion of the filtered clusters are also deemed as invalid by human. In Table 13, we show the effect of using different models in the self-verification module, where we find larger models, such as GPT-4 and GPT-4o provide better verification.

Table 11: Ablation study of the proposed LLM cluster metric.

(a) K

| $K$ | Score |
|-----|-------|
| 5 | $5.12_{\pm 0.14}$ |
| 10 | $3.99_{\pm 0.05}$ |
| 15 | $3.48_{\pm 0.29}$ |
| 20 | $3.13_{\pm 0.46}$ |
| 50 | $2.05_{\pm 0.83}$ |
| 100 | $1.49_{\pm 1.02}$ |

(b) N

| $N$ | Score |
|-----|-------|
| 100 | $4.15_{\pm 1.38}$ |
| 1000 | $3.71_{\pm 0.25}$ |
| 5000 | $3.99_{\pm 0.05}$ |
| 10000 | $4.02_{\pm 0.03}$ |

(c) Component

| Component | Score |
|-----------|-------|
| only clustering | $2.67_{\pm 0.46}$ |
| w/o verification | $3.74_{\pm 0.63}$ |
| whole | $3.99_{\pm 0.05}$ |

(d) Model

| Model | Score |
|-------|-------|
| GPT-3.5 | $3.83_{\pm 0.11}$ |
| GPT-4 | $3.99_{\pm 0.05}$ |
| GPT-4o | $3.92_{\pm 0.14}$ |
| Llama-3.1 | $3.76_{\pm 0.28}$ |

Table 12: Human evaluation on the filtered clusters from the self-verification module.

| Topic/#Samples | Clusters | Self-verified Invalid Clusters | Human-verified Invalid Clusters |
|----------------|----------|--------------------------------|---------------------------------|
| 100/10 | 12943 | 248 | 221 |
| 100/20 | 15216 | 350 | 329 |

## C  DIVERSITY METRICS

**Context Length** refers to the average length of the sequences in the dataset. Longer contexts can indicate more complex data structures and richer narratives. By analyzing context length, we can infer the ability of the synthetic data to capture long-term dependencies and intricate patterns.

**Self-repetition Score** quantifies how often sequences or phrases are repeated within the dataset. Lower scores suggest higher diversity, as the model generates more varied outputs rather than reiterating the same phrases. High self-repetition can indicate overfitting or a lack of creativity in the synthetic generation process.

**N-gram Diversity Score** measures the variability of contiguous sequences of 'n' items in the dataset. By examining different 'n' values (e.g., unigrams, bigrams, trigrams), this score highlights how varied the generated text is at multiple granularities. A higher N-gram diversity score indicates more creative and less predictable outputs, which is often desirable in synthetic data generation.

**Compression Ratio** assesses the dataset's redundancy by compressing it and comparing the compressed size to the original size. A lower compression ratio suggests that the data is less repetitive and more diverse. This metric provides a quantitative way to gauge the amount of unique information within the dataset.

**Perplexity** is a measure of how well a probability model predicts a sample. In the context of synthetic data, lower perplexity indicates that the model can predict the data more confidently, which may imply less diversity if the model is overconfident. Higher perplexity, conversely, can indicate that the model encounters more unexpected or varied data, pointing towards greater diversity.

**Perplexity Gap** measures the difference in perplexity between GPT-2-L and GPT-2-XL (Radford et al., 2019), used to assess dataset diversity. A smaller gap indicates less diversity, while a larger gap reflects greater variability and complexity in the data.

**K-means Clustering** is used to partition the dataset into distinct groups based on feature similarity. By analyzing the number and distribution of clusters, we can gain insights into the inherent diversity of the data. However, traditional clustering methods like K-means may struggle with high-dimensional, complex data structures, often oversimplifying the richness of the data.

## D  LLM CLUSTERING

In this section, we provide detailed prompt templates, prompt examples, and output examples of the proposed LLM Cluster-agent metric. The prompt templates we used include **metadata and metric generation**, **metadata and metric summary**, **high-level criteria definition summary**, **clustering**, and **self-verification**.

Table 13: Ablation on the models used in self-verification module.

| Self-Verification Model | Invalid Clusters | Cluster Score |
|---|---|---|
| GPT-4o | 248 | 3.99 |
| GPT-4 | 254 | 4.03 |
| GPT-3.5 | 218 | 3.81 |
| Llama-3.1 | 192 | 3.65 |

## D.1   PROMPTS TEMPLATES IN PIPELINE

---
**Metadata and Metric Generation Prompt Template**

```
# Task
You are going to evaluate the diversity of text corpus based on clustering.  Before clustering,
your task is to come up with a set of cluster metadata and cluster metrics that can measure the true
underlying diversity, better group samples, and better discriminate between clusters.

## Instructions
To design the metadata and metrics, you will be given a set of individual samples, and return 3-5
metadata and 3-5 metrics and their definitions that can help better cluster them.  You should avoid
generic terms for metadata and metrics as they are not suitable for fine-grained clustering.  I will
run this for multiple rounds and gather the unique metadata and metrics eventually.

## Outputs Demonstration and Format
Your output needs to be in the following JSON format:
```json
{{
'metadata': {{
# [a dict of 3-5 metadata]
'metadata_name': "concrete definition of metadata name, use hierarchy to if necessary (level 1/level
2/level 3/.../level k), where each level is more nuanced.",
..., }}
'metric': {{
# [a dict of 3-5 metrics]
'metric_name': "specific justification and analysis for metric that will be used for clustering.  You
need define detailed scoring from 1-5 for each metric",
..., }}
}}
```

## All samples
samples

## Outputs
```

## Metadata Summary Prompt Template

```
# Task
Your tasks is to group a dictionary of metadata and their definition that describes the
characteristics of a group of sampled texts.  You need to summarize and return **K=k** metadata
and their unique definition, which will be used later to cluster the text data.  The metadata needs
be able to measure the true underlying diversity, better group samples, and better discriminate
between clusters.

## Instructions
The metadata dictionary has the following structure:
```
{{
'metadata_1': [ 'definition_1', 'definition_2', ... ],
'metadata_2': [ 'definition_1', 'definition_2', ... ],
...
}}
```
Each key in the dictionary indicates a unique metadata and each item indicates the list of definition
of this metadata (generated by different round of samples) You need first to collect all unique
keys according to their meaning and definition, and choose and summarize them as the general ones.
Then you need to refine the definition for each unique key to make it **concrete** and **suitable
to cluster** the data.  There might be more than 5 keys in the dictionary and you need to summarize
them.

## Outputs Demonstration and Format
Your output needs to be in the following JSON format:
```json
{{
'metadata_1': 'definition of metadata_1, use hierarchy levels along with definition if necessary (as
level1/level2/level3...), where deeper levels are more nuanced',
'metadata_2': 'definition of metadata_2, use hierarchy levels along with definition if necessary (as
level1/level2/level3...), where deeper levels are more nuanced',
...
'metadata_k': 'definition of metadata_k, use hierarchy levels along with definition if necessary (as
level1/level2/level3...), where deeper levels are more nuanced',
}}
```

## All metadata
{metadata}

## Outputs
```

## Metric Summary Prompt Template

```
# Task
Your tasks is to group a dictionary of metrics and their definition that measures the key
characteristics of a group of sampled texts.  You need to summarize and return **K=k** metrics
and their unique definition and score levels (from 1-5) that will be used later to cluster the text
data, so the metrics needs be able to measure the true underlying diversity, better group samples,
and better discriminate between clusters.

## Instructions
The metric dictionary has the following structure:
```
{{
'metric_1': ['definition_1', 'definition_2', ...],
'metric_2': ['definition_1', 'definition_2', ...],
...
}}
```
Each key in the dictionary indicates a unique metric and each item indicates the list of definition
of this metric (generated by different round of samples) You need first to collect all unique keys
according to their meaning and definition, and choose and summarize them as the general ones.  Then
you need to refine the definition for each unique key to make it **concrete** and **suitable to
cluster and score** the data.  There might be more than 5 keys in the dictionary and you need to
summarize them.

## Outputs Demonstration and Format
Your output needs to be in the following JSON format:
```json
{{
'metric_1': 'definition of metric_1, score 1-5 definition',
'metric_2': 'definition of metric_2, score 1-5 definition,
...
'metric_k': 'definition of metric_k, score 1-5 definition'
}}
```

## All metadata
{metric}

## Outputs
```

---

**Criteria Summary Prompt Template**

```
# Task
Given a group of metadata and metrics with their definitions, your task is to summarize each metadata
and metric concisely as one sentence, which will be used as criteria guidance for clustering the text
data.

## Instructions
The metadata and metric dictionary have the following structure:
```
{{
'metadata_1/metric_1': 'definition of metadata_1/metric_1',
'metadata_2/metric_2': 'definition of metadata_2/metric_2'
...
}}
```

## Outputs Demonstration and Format
Your output needs to be in the following JSON format:
```json
{{
'metadata_1': 'concise criteria for clustering text samples based on definition of metadata_1', ...
'metadata_k': 'concise criteria for clustering text samples based on definition of metadata_k',
'metric_1': 'concise criteria for clustering text samples based on definition of metric_1', ...
'metric_2': 'concise criteria for clustering text samples based on definition of metric_2',
}}
```
You need to summarize the criteria from the definition of each metric and metadata to make it a
concise guidance for clustering text.

## Metadata
metadata

## Metric
metric

## Outputs
```

---

**Clustering Prompt Template**

```
# Task
You are evaluating the diversity of synthetic data.  Given a set of randomly sampled synthetic text
from the dataset, your task is to measure the absolute diversity of these samples.

## Instructions To measure the diversity, you need to cluster the samples by a set of metrics and
metadata.

## Clustering Criteria:  1.  {metadata_1}:  {criteria definition of metadata_1 }
2.  {metric_1}:  {criteria definition of metric_1 }
...
2n-1.  {metadata_n}:  {criteria definition of metadata_n }
2n.  {metric_n}:  {criteria definition of metric_n }

## Clusters
You need to output all the clusters from the given samples, even if a cluster contains only one
sample.  Your output needs to be in the following JSON format:
'''json
{{
"clusters":  [
{{
"cluster":  n,
"sample indices":  [sample indices in the cluster],
"uniqueness reasoning":  "justification of what makes this group/cluster unique, how is it different
than the other clusters as a group",
"cluster_metadata":
{{
"metadata_1":  "definition of metadata_1",
...
}},
"cluster_metrics":
{{
"metric_1":
{{ "reasoning":  "definition of this metric and its score definition",
"score":  int 5-1 score
}},
...
}}
}},
...
]
...
}}
'''

## All samples
{samples}

## Outputs
```

**Self-Verification Prompt Template**

```
# Task
You are measuring the diversity of text data. Given a set of text samples and a set of dictionary of
clustered text indices with corresponding reasoning over text metadata and metrics, your task is to
verify whether the clustered text samples can be clustered as a group. The verification should be
based on the similarity of the text samples, and the reasoning part from the cluster dictionary.

## Illustration
You will be given a set of samples:
```
1. Text 1
2. Text 2
...
K. Text k
```
and a set of dictionary of clusters:
```
[
{{
'cluster': 1,
'sample indices': [...],
'reasoning': ...
}},
...
]
```
Your task is to verify whether each cluster is reasonable and return a binary indication 0/1 for each
cluster as:
```
[
{{ 'cluster': 1, 'valid': 0/1, 'reasoning':...}},
...
]
```
where 0 indicates an invalid cluster and 1 indicate a valid cluster. You should include your
detailed reasoning for the validation each cluster, e.g., these samples can be clustered together
as they all follow the same topic, or these samples cannot be clustered because of their difference.
You should mark all clusters with one single sample as 1.

## Samples
{samples}

## Clusters
{clusters}

## Outputs
```

## D.2   EXAMPLES OF PROMPTING OUTPUTS IN PIPELINE

**Metadata and Metric Generation Example Output**

```
{"metadata": {"content_complexity_level": "The depth and sophistication of content, ranging from
basic definitions (level 1) to advanced theoretical applications and real-world implications (level
5).", "disciplinary_focus": "The primary academic or professional discipline the content pertains to,
from general knowledge (level 1) to highly specialized subfields (level 5).", "terminology_density":
"The frequency and distribution of specialized terms and jargon within the text, measured from common
language (level 1) to dense technical language (level 5).", "conceptual_novelty": "The degree of
innovation or rarity of the concepts presented, from widely understood (level 1) to cutting-edge
or groundbreaking (level 5).", "argumentation_structure": "The organization and presentation of
arguments or assertions, from simple (level 1) to highly complex and multi-layered (level 5)."},
"metric": {"interdisciplinary_citation_frequency": "The rate at which content references or draws
upon knowledge from other disciplines, scored from isolated (1) to highly interdisciplinary (5).",
"conceptual_coherence_score": "The internal consistency and logical flow of concepts, rated from
fragmented (1) to tightly integrated (5).", "novelty_impact_factor": "The potential of the content to
contribute new insights or shifts in understanding, scored from minimal (1) to transformative (5).",
"jargon_comprehension_load": "The cognitive load required to understand the specialized language
used, measured from light (1) to heavy (5).", "argumentative_density": "The richness and complexity
of the reasoning presented, from sparse (1) to dense (5)."} }
```

### Metadata Summary Example Output

```
{"Subject Domain": "The specific academic or professional field to which the sample text is related,
indicative of the specialized content domain (e.g., Medical/Neurology, Engineering/Mechanical,
Humanities/Philosophy).", "Conceptual Density": "The frequency and complexity of specialized
concepts and terminology within the text, indicating the depth of knowledge required to understand
the content, with levels denoting richness (Level 1: Sparse – Level 5: Extremely Dense).",
"Temporal Relevance": "The extent to which the text content is historically grounded or
forward-looking, categorized as Historical/Current/Future-Oriented, crucial for clustering based
on time relevance.", "Narrative Structure": "The organization and flow of the content, ranging
from linear and chronological to complex and non-linear, including the presence of argumentative
frameworks or narrative arcs."}
```

### Metric Summary Example Output

```
{"Conceptual Clarity": "Evaluates the clarity and precision with which core concepts are presented
in the text. Definitions and explanations should be concise and easily understandable, with a score
of 1 indicating vague and confusing explanations, and a score of 5 indicating exceptional clarity
and ease of understanding for the intended audience.", "Interdisciplinary Integration": "Assesses
the degree to which a text integrates knowledge from multiple disciplines, demonstrating the text's
breadth and depth. A score of 1 indicates content with a single-discipline focus, while a score of
5 indicates content that is highly interdisciplinary, weaving together multiple fields seamlessly.",
"Information Density": "Measures the quantity and significance of information conveyed per unit
of text. A score of 1 indicates sparse or superficial details, while a score of 5 indicates a text
that is rich in detail and has significant depth, covering both the breadth and depth of content.",
"Lexical Diversity": "Analyzes the variety of vocabulary used in the text, providing insight into
the text's linguistic complexity. A score of 1 indicates low diversity with repetitive use of common
words, while a score of 5 indicates high diversity with a wide range of advanced and specialized
terms."}
```

### Criteria Summary Example Output

```
{"Subject Domain": "Cluster text samples based on their specific academic or professional field.",
"Conceptual Density": "Group text by the level of specialized concepts and terminology, from sparse
to extremely dense.", "Temporal Relevance": "Organize text content by its historical grounding
or orientation towards the current or future.", "Narrative Structure": "Cluster texts by the
organization of content from linear to complex and presence of narrative elements.", "Conceptual
Clarity": "Sort texts based on how clearly and precisely core concepts are presented, from vague to
exceptionally clear.", "Interdisciplinary Integration": "Cluster texts by the extent of knowledge
integration from multiple disciplines, from single-discipline to highly interdisciplinary.",
"Information Density": "Group texts by the quantity and significance of information, from sparse
to rich in detail.", "Lexical Diversity": "Organize texts according to vocabulary variety, from low
to high lexical diversity."}
```

## Clustering Example Output

```
{ "clusters": [
{"cluster": 1, "sample indices": [5], "uniqueness reasoning": "This group is unique due to its
focus on the historical aspect of organized crime, which is less conceptually dense, with a temporal
relevance rooted in a past event, and has a linear narrative structure.", "cluster metadata":
{"subject domain": "History/Criminology", "conceptual density": "Level 2: Moderate", "temporal
relevance": "Historical", "narrative structure": "Linear"}, "cluster metrics": {"conceptual
clarity": {"reasoning": "Concepts such as 'Consigliere' and 'French Connection' are explained
clearly, making the text understandable to a general audience.", "score": 4}, "interdisciplinary
integration": {"reasoning": "The text focuses on a specific historical event without incorporating
multiple disciplines.", "score": 1}, "information density": {"reasoning": "Information is
presented in a focused manner, without extensive detail on broader implications or contexts.",
"score": 2}, "lexical diversity": {"reasoning": "The vocabulary used is moderately varied, with
some industry-specific terms like 'Consigliere'.", "score": 3}}},

{"cluster": 2, "sample indices": [1, 7], "uniqueness reasoning": "This cluster is characterized by
its highly specialized subject matter in medicine and oncology, with dense conceptual content and a
clear focus on current and future treatment strategies.", "cluster metadata": {"subject domain":
"Medical/Oncology", "conceptual density": "Level 5: Extremely Dense", "temporal relevance":
"Current/Future-Oriented", "narrative structure": "Complex"}, "cluster metrics": {"conceptual
clarity": {"reasoning": "The text includes specialized medical terminology and concepts
that require a higher level of understanding.", "score": 3}, "interdisciplinary integration":
{"reasoning": "The sample integrates knowledge from multiple disciplines within medicine, such as
immunotherapy and genetics.", "score": 4}, "information density": {"reasoning": "The text is rich
in detail, covering the intricacies of cancer treatment and immunological concepts.", "score": 5},
"lexical diversity": {"reasoning": "The text uses a high variety of specialized terms, indicative
of a high lexical diversity.", "score": 5}}}, ... ] }
```

## Self-Verification Example Output

```
"validation": [ {"cluster": 1, "valid": 0, "reasoning": "The samples in Cluster 1 are from vastly
different scientific domains - one discusses Earth's magnetic properties (geology) while the other
is about viral RNA synthesis (virology). Despite both being in natural sciences, the topics are
too distinct without any significant overlap in content or terminology to be clustered effectively.
The reasoning provided does not justify the clustering based on mere categorization within natural
sciences; it lacks a cohesive thematic or conceptual link."},
{"cluster": 2, "valid": 1, "reasoning": "Cluster 2 samples share a common theme of practical
applications in their respective fields, focusing on processes, strategies, and advancements. Sample
3 discusses data management, Sample 5 focuses on advancements in materials for devices with practical
applications, and Sample 9 talks about environmental impacts in journalism. Although the fields are
different, the underlying commonality in their practical and solution-oriented narrative justifies
the clustering."},
{"cluster": 3, "valid": 0, "reasoning": "Samples in Cluster 3 cover a wide range of specialized
knowledge across various fields including physics, philosophy, chemistry, and engineering. While
each sample is dense with information and specialized terminology, the subject matters are too
diverse to be clustered together effectively. For instance, queer philosophy (Sample 6) and
structural engineering (Sample 10) do not share a common conceptual or thematic foundation that
would warrant their clustering beyond their academic complexity."} ], }
```

# E    SEEDING TOPICS OF SYNTHETIC GENERATION

We provide more details on the seeding topics that we used for synthetic data generation.

## E.1 EXAMPLES OF TOPIC SEEDS

---

**Topic Seeds Example**

```
"Physical Sciences/Quantum physics/Degenerate quantum gases and atom optics/Rydberg atoms and ions
and quantum information/quantum memory and communication":  [
"Atom Optics",
"Boson Sampling",
"Cavity Quantum Electrodynamics",
"Collisional Blockade",
"Degenerate Quantum Gases",
"Dipole Blockade",
"Fock State",
"Frequency Combs",
"Isotope Shift",
"Jaynes-Cummings Model",
"Magneto-optical Traps",
"Many-body Systems",
...
]
"Engineering/Chemical engineering/Wastewater treatment processes/Resource recovery and circular
economy/Water reclamation and reuse":  [
"Advanced Oxidation Process",
"Bacterial Oxidation",
"Biosolids",
"Blackwater",
"Chemical Precipitation",
"Combined Sewer Overflow",
"Contaminants of Emerging Concern",
"Decentralized Wastewater Treatment",
"Dissolved Air Flotation",
"Electrocoagulation",
"Greywater",
"Heavy Metals Removal",
"Hydraulic Retention Time",
...
]
"Human Society/Sociology/Sociology of religion/Religion and Culture/Religion and transnationalism and
migration":  [
"Adventists",
"African Diaspora",
"Aliyah",
...
]
```

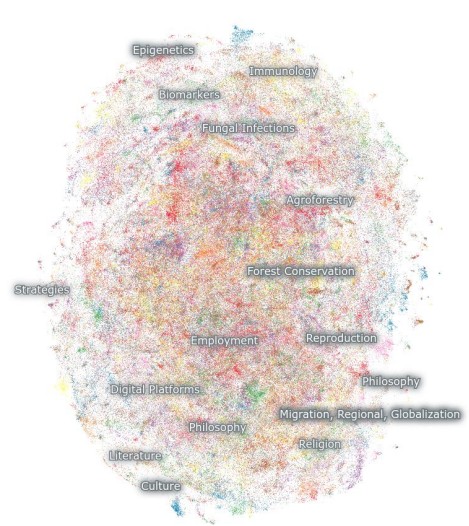

Figure 12: Visualization on the clustering of topic seeds.

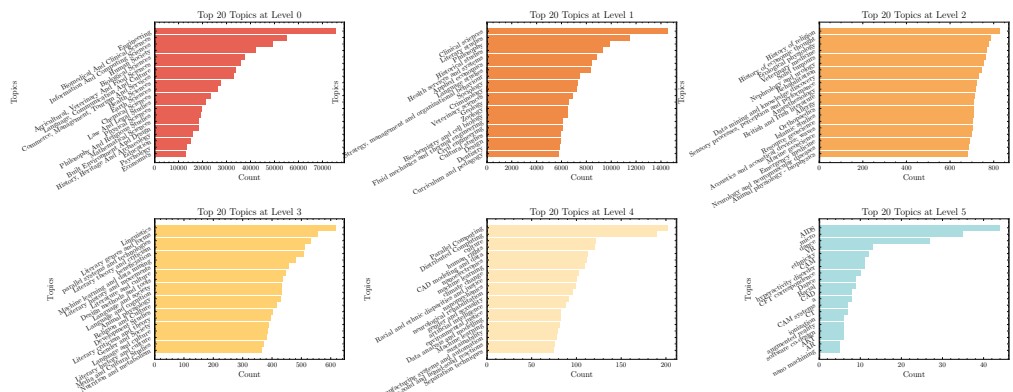

Figure 13: Distribution of top-20 topics at each hierarchical level.

## E.2 VISUALIZATION OF THE TOPIC SEEDS

# F SYNTHETIC DATA GENERATION

## F.1 GENERATION PROMPT TEMPLATES

### F.1.1 *Topic*

---

**_Topic_ Prompt Template**

```
# Task
Generate consecutive passages in textbook style, utilizing the following instructions.

## Instructions
- Assume the reader already has a basic knowledge of the high-level topic {topic}, but they are
looking to learn more about subtopics including {subtopic}.
- Generate 3-5 consecutive passages exploring the subject, increasing in nuance and detail by
passage, by that, I mean, increase the detail and example use of what the reader might learn from
the text.
- For each passage, you can select from the list of relevant keywords to guide the content of the
passages.
- Lastly, follow up the passages with a multiple choice question to test the most complex ideas in
learned from the passages, this will serve as a tool for the reader to test what they have learned
from this textbook.

## Topic
{topic}

## Subtopic
{subtopic}

## Keyword
{keyword}

## Output
- Your output must be in the following JSON format:
{{ "passages": [ {{
"nuanced_content_to_be_learned": [keyword style list of new and intellectually complex concepts
learned in this passage],
"passage": "The passage text goes here."
}},
.... ],
"multiple_choice_question": {{
"question": "MC question utilizing the complex ideas learned in the passages.",
"options": ["Option 1", "Option 2", "Option 3", "Option 4"] (do not use any indexing),
"answer_label": "The correct answer label. Return the exact text from options"
"step_by_step_answer_explanation": "a detailed step-by-step layout of how one arrives at this answer
and what relevant information from the passages led to this answer."
}}
}}
```

## F.1.2 *Topic Styles*

---

**Topic Textbook Narrative Prompt Template**

```
# Task
Generate consecutive passages in an narrative textbook style, utilizing the following instructions.

## Instructions
- Write an extensive and detailed course unit suitable for a textbook.
- Assume the reader already has a basic knowledge of the high-level topic {topic}, but they are
looking to learn more about subtopics including {subtopic}.
- Do not just list concepts, but develop each one in detail before moving to the next, as we
prioritize depth of understanding and comprehensive exploration of the subject matter over breadth.
- Engagement:  Use a narrative style akin to Michael Lewis, making it captivating and
thought-provoking.
- Relevance:  Connect the topic with current trends, real-life examples, or recent studies.  Do not
use images.
- Generate 3-5 consecutive passages exploring the subject, increasing in nuance and detail by
passage, by that, I mean, increase the detail and example use of what the reader might learn from
the text.
- For each passage, you can select from the list of relevant keywords to guide the content of the
passages.
- Lastly, follow up the passages with a multiple choice question to test the most complex ideas in
learned from the passages, this will serve as a tool for the reader to test what they have learned
from this textbook.
Do not include a title or an introduction, simply write the content without headlines and
introductory phrases.  Do not use images.

## Topic
{topic}

## Subtopic
{subtopic}

## Keyword
{keyword}
## Output
- Your output must be in the following JSON format:
{{ "passages":  [ {{
"nuanced_content_to_be_learned":  [keyword style list of new and intellectually complex concepts
learned in this passage],
"passage":  "The passage text goes here."
}},
.... ],
"multiple_choice_question":  {{
"question":  "MC question utilizing the complex ideas learned in the passages.",
"options":  ["Option 1", "Option 2", "Option 3", "Option 4"] (do not use any indexing),
"answer_label":  "The correct answer label.  Return the exact text from options"
"step_by_step_answer_explanation":  "a detailed step-by-step layout of how one arrives at this answer
and what relevant information from the passages led to this answer."
}}
}}
```

---

**_Topic Textbook Academic_ Prompt Template**

```
# Task
Generate consecutive passages in an academic textbook style, utilizing the following instructions.

## Instructions
- Write an extensive and detailed course unit suitable for a textbook targeted at college students.
- Assume the reader already has a basic knowledge of the high-level topic {topic}, but they are
looking to learn more about subtopics including {subtopic}.
- Engagement:  Write with an academic, professional and engaging tone that captivates interest.
- Application:  Incorporate specific, practical examples, such as proofs in calculus or critical
dates and figures in history.
- Generate 3-5 consecutive passages exploring the subject, increasing in nuance and detail by
passage, by that, I mean, increase the detail and example use of what the reader might learn from
the text.
- For each passage, you can select from the list of relevant keywords to guide the content of the
passages.
- Lastly, follow up the passages with a multiple choice question to test the most complex ideas in
learned from the passages, this will serve as a tool for the reader to test what they have learned
from this textbook.
Do not include a title or an introduction, simply write the content without headlines and
introductory phrases.  Do not use images.

## Topic
{topic}

## Subtopic
{subtopic}

## Keyword
{keyword}

## Output
- Your output must be in the following JSON format:
{{ "passages":  [ {{
"nuanced_content_to_be_learned":  [keyword style list of new and intellectually complex concepts
learned in this passage],
"passage":  "The passage text goes here."
}},
.... ],
"multiple_choice_question":  {{
"question":  "MC question utilizing the complex ideas learned in the passages.",
"options":  ["Option 1", "Option 2", "Option 3", "Option 4"] (do not use any indexing),
"answer_label":  "The correct answer label.  Return the exact text from options"
"step_by_step_answer_explanation":  "a detailed step-by-step layout of how one arrives at this answer
and what relevant information from the passages led to this answer."
}}
}}
```

---

### *Topic Blogpost* Prompt Template

```
# Task
Generate consecutive passages in a blog post style, utilizing the following instructions.
## Instructions
- Write an informative and insightful blog post that expands upon the topic {topic}.
- Assume the reader already has a basic knowledge of the high-level topic {topic}, but they are
looking to learn more about subtopics including {subtopic}.
- Generate 3-5 consecutive passages exploring the subject, increasing in nuance and detail by
passage, by that, I mean, increase the detail and example use of what the reader might learn from
the text.
- For each passage, you can select from the list of relevant keywords to guide the content of the
passages.
- Your post should delve into the nuances of the topic, offering fresh perspectives and deeper
analysis.
- Inform:  Provide valuable, well-researched information that educates the reader.
- Engage:  Write in a conversational tone that connects with the audience, making complex ideas
accessible.
- Illustrate:  Use examples, anecdotes, or personal experiences to bring the topic to life.
- Lastly, follow up the passages with a multiple choice question to test the most complex concepts in
learned from the passages, this will serve as a tool for the reader to test what they have learned
from this blog post.
Do not give a title and do not start with sentences like "Have you ever..." or "Hello dear
readers..", simply write the content without these introductory phrases.

## Topic
{topic}

## Subtopic
{subtopic}

## Keyword
{keyword}

## Output
- Your output must be in the following JSON format:
{{ "passages":  [ {{
"nuanced_content_to_be_learned":  [keyword style list of new and intellectually complex concepts
learned in this passage],
"passage":  "The passage text goes here."
}},
....  ],
"multiple_choice_question":  {{
"question":  "MC question utilizing the complex ideas learned in the passages.",
"options":  ["Option 1", "Option 2", "Option 3", "Option 4"] (do not use any indexing),
"answer_label":  "The correct answer label.  Return the exact text from options"
"step_by_step_answer_explanation":  "a detailed step-by-step layout of how one arrives at this answer
and what relevant information from the passages led to this answer."
}}
}}
```

*Topic Wikihow* Prompt Template

```
# Task
Generate consecutive passages in a Wikihow style, utilizing the following instructions.
## Instructions
- Write a long and very detailed tutorial that could be part of WikiHow.
- Assume the reader already has a basic knowledge of the high-level topic {topic}, but they are
looking to learn more about subtopics including {subtopic}.
- Generate 3-5 consecutive passages exploring the subject, increasing in nuance and detail by
passage, by that, I mean, increase the detail and example use of what the reader might learn from
the text.
- For each passage, you can select from the list of relevant keywords to guide the content of the
passages.
- Include in depth explanations for each step and how it helps achieve the desired outcome, inluding
key tips and guidelines.
- Ensure clarity and practicality, allowing readers to easily follow and apply the instructions.  Do
not use images.,
- Lastly, follow up the passages with a multiple choice question to test the most complex concepts in
learned from the passages, this will serve as a tool for the reader to test what they have learned
from this WikiHow.
Do not include a title or an introduction, simply write the content without headlines and
introductory phrases.  Do not use images.

## Topic
{topic}

## Subtopic
{subtopic}

## Keyword
{keyword}

## Output
- Your output must be in the following JSON format:
{{ "passages":  [ {{
"nuanced_content_to_be_learned":  [keyword style list of new and intellectually complex concepts
learned in this passage],
"passage":  "The passage text goes here."
}},
.... ],
"multiple_choice_question":  {{
"question":  "MC question utilizing the complex ideas learned in the passages.",
"options":  ["Option 1", "Option 2", "Option 3", "Option 4"] (do not use any indexing),
"answer_label":  "The correct answer label.  Return the exact text from options"
"step_by_step_answer_explanation":  "a detailed step-by-step layout of how one arrives at this answer
and what relevant information from the passages led to this answer."
}}
}}
```

F.1.3  *Topic Styles Persona*

---

**Topic Textbook Narrative Persona** Prompt Template

```
# Task
Generate consecutive passages in a narrative textbook style, utilizing the following instructions.

## Instructions
- Write an extensive and detailed course unit suitable for a textbook targeted at specified persona.
You will be given a list of persona and need to select the most suitable one for the content
generation.
- Assume the reader already has a basic knowledge of the high-level topic {topic}, but they are
looking to learn more about subtopics including {subtopic}.
- Do not just list concepts, but develop each one in detail before moving to the next, as we
prioritize depth of understanding and comprehensive exploration of the subject matter over breadth.
- Engagement:  Use a narrative style akin to Michael Lewis, making it captivating and
thought-provoking.
- Relevance:  Connect the topic with current trends, real-life examples, or recent studies.  Do not
use images.
- Generate 3-5 consecutive passages exploring the subject, increasing in nuance and detail by
passage, by that, I mean, increase the detail and example use of what the reader might learn from
the text.
- For each passage, you can select from the list of relevant keywords to guide the content of the
passages.
- Lastly, follow up the passages with a multiple choice question to test the most complex ideas in
learned from the passages, this will serve as a tool for the reader to test what they have learned
from this textbook.

## Topic
{topic}

## Subtopic
{subtopic}

## Keyword
{keyword}

## Persona
{persona}

## Output
- Your output must be in the following JSON format:
{{ "passages":  [ {{
"nuanced_content_to_be_learned":  [keyword style list of new and intellectually complex concepts
learned in this passage],
"passage":  "The passage text goes here."
}},
.... ],
"multiple_choice_question":  {{
"question":  "MC question utilizing the complex ideas learned in the passages.",
"options":  ["Option 1", "Option 2", "Option 3", "Option 4"] (do not use any indexing),
"answer_label":  "The correct answer label.  Return the exact text from options"
"step_by_step_answer_explanation":  "a detailed step-by-step layout of how one arrives at this answer
and what relevant information from the passages led to this answer."
}}
}}
```

## *Topic Textbook Academic Persona* Prompt Template

```
# Task
Generate consecutive passages in an academic textbook style, utilizing the following instructions.

## Instructions
- Write an extensive and detailed course unit suitable for a textbook targeted at specified persona.
You will be given a list of persona and need to select the most suitable one for the content
generation.
- Assume the reader already has a basic knowledge of the high-level topic {topic}, but they are
looking to learn more about subtopics including {subtopic}.
- Engagement: Write with an academic, professional and engaging tone that captivates interest.
- Application: Incorporate specific, practical examples, such as proofs in calculus or critical
dates and figures in history.
- Generate 3-5 consecutive passages exploring the subject, increasing in nuance and detail by
passage, by that, I mean, increase the detail and example use of what the reader might learn from
the text.
- For each passage, you can select from the list of relevant keywords to guide the content of the
passages.
- Lastly, follow up the passages with a multiple choice question to test the most complex ideas in
learned from the passages, this will serve as a tool for the reader to test what they have learned
from this textbook.
Do not include a title or an introduction, simply write the content without headlines and
introductory phrases. Do not use images.

## Topic
{topic}

## Subtopic
{subtopic}

## Keyword
{keyword}

## Persona
{persona}

## Output
- Your output must be in the following JSON format:
{{ "passages": [ {{
"nuanced_content_to_be_learned": [keyword style list of new and intellectually complex concepts
learned in this passage],
"passage": "The passage text goes here."
}},
.... ],
"multiple_choice_question": {{
"question": "MC question utilizing the complex ideas learned in the passages.",
"options": ["Option 1", "Option 2", "Option 3", "Option 4"] (do not use any indexing),
"answer_label": "The correct answer label. Return the exact text from options"
"step_by_step_answer_explanation": "a detailed step-by-step layout of how one arrives at this answer
and what relevant information from the passages led to this answer."
}}
}}
```

*Topic Blogpost Persona* Prompt Template

```
# Task
Generate consecutive passages in a blog post style, utilizing the following instructions.
## Instructions
- Write an informative and insightful blog post targeted at specified persona.  You will be given a
list of persona and need to select the most suitable one for the content generation.
- Assume the reader already has a basic knowledge of the high-level topic {topic}, but they are
looking to learn more about subtopics including {subtopic}.
- Generate 3-5 consecutive passages exploring the subject, increasing in nuance and detail by
passage, by that, I mean, increase the detail and example use of what the reader might learn from
the text.
- For each passage, you can select from the list of relevant keywords to guide the content of the
passages.
- Your post should delve into the nuances of the topic, offering fresh perspectives and deeper
analysis.
- Inform:  Provide valuable, well-researched information that educates the reader.
- Engage:  Write in a conversational tone that connects with the audience, making complex ideas
accessible.
- Illustrate:  Use examples, anecdotes, or personal experiences to bring the topic to life.
- Lastly, follow up the passages with a multiple choice question to test the most complex concepts in
learned from the passages, this will serve as a tool for the reader to test what they have learned
from this blog post.  Do not give a title and do not start with sentences like "Have you ever..." or
"Hello dear readers..", simply write the content without these introductory phrases.

## Topic
{topic}

## Subtopic
{subtopic}

## Keyword
{keyword}

## Persona
{persona}

## Output
- Your output must be in the following JSON format:
{{ "passages":  [ {{
"nuanced_content_to_be_learned":  [keyword style list of new and intellectually complex concepts
learned in this passage],
"passage":  "The passage text goes here."
}},
.... ],
"multiple_choice_question":  {{
"question":  "MC question utilizing the complex ideas learned in the passages.",
"options":  ["Option 1", "Option 2", "Option 3", "Option 4"] (do not use any indexing),
"answer_label":  "The correct answer label.  Return the exact text from options"
"step_by_step_answer_explanation":  "a detailed step-by-step layout of how one arrives at this answer
and what relevant information from the passages led to this answer."
}}
}}
```

---

*Topic Wikihow Persona* Prompt Template

```
# Task
Generate consecutive passages in a Wikihow style, utilizing the following instructions.
## Instructions
- Write a long and very detailed tutorial that could be part of WikiHow targeted at specified
persona. You will be given a list of persona and need to select the most suitable one for the
content generation.
- Assume the reader already has a basic knowledge of the high-level topic {topic}, but they are
looking to learn more about subtopics including {subtopic}.
- Generate 3-5 consecutive passages exploring the subject, increasing in nuance and detail by
passage, by that, I mean, increase the detail and example use of what the reader might learn from
the text.
- For each passage, you can select from the list of relevant keywords to guide the content of the
passages.
- Include in depth explanations for each step and how it helps achieve the desired outcome, inluding
key tips and guidelines.
- Ensure clarity and practicality, allowing readers to easily follow and apply the instructions. Do
not use images.,
- Lastly, follow up the passages with a multiple choice question to test the most complex concepts in
learned from the passages, this will serve as a tool for the reader to test what they have learned
from this WikiHow. Do not include a title or an introduction, simply write the content without
headlines and introductory phrases.

## Topic
{topic}

## Subtopic
{subtopic}

## Keyword
{keyword}

## Persona
{persona}

## Output
- Your output must be in the following JSON format:
{{ "passages": [ {{
"nuanced_content_to_be_learned": [keyword style list of new and intellectually complex concepts
learned in this passage],
"passage": "The passage text goes here."
}},
.... ],
"multiple_choice_question": {{
"question": "MC question utilizing the complex ideas learned in the passages.",
"options": ["Option 1", "Option 2", "Option 3", "Option 4"] (do not use any indexing),
"answer_label": "The correct answer label. Return the exact text from options"
"step_by_step_answer_explanation": "a detailed step-by-step layout of how one arrives at this answer
and what relevant information from the passages led to this answer."
}}
}}
```

### F.1.4 *Multi-Topic Styles Persona*

---

**Multi-Topic Textbook Narrative Persona Prompt Template**

```
# Task
Generate consecutive passages in a narrative textbook style, utilizing the following instructions.

## Instructions
- Write an extensive and detailed course unit suitable for a textbook targeted at specified persona.
You will be given a list of persona and need to select the most suitable one for the content
generation.
- You will be given a list of topics and subtopics for each topic.  You need combine the suitable
topics and subtopics for the content generation.  If there is no suitable combination, just use one
topic and all of its subtopics.
- Assume the reader already has a basic knowledge of the high-level topic, but they are looking to
learn more about subtopics.
- Do not just list concepts, but develop each one in detail before moving to the next, as we
prioritize depth of understanding and comprehensive exploration of the subject matter over breadth.
- Engagement:  Use a narrative style akin to Michael Lewis, making it captivating and
thought-provoking.
- Relevance:  Connect the topic with current trends, real-life examples, or recent studies.  Do not
use images.
- Generate 3-5 consecutive passages exploring the subject, increasing in nuance and detail by
passage, by that, I mean, increase the detail and example use of what the reader might learn from
the text.
- For each passage, you can select from the list of relevant keywords to guide the content of the
passages.
- Lastly, follow up the passages with a multiple choice question to test the most complex ideas in
learned from the passages, this will serve as a tool for the reader to test what they have learned
from this textbook.

## Topic
{topic}

## Subtopic
{subtopic}

## Keyword
{keyword}

## Persona
{persona}

## Output
- Your output must be in the following JSON format:
{{ "passages":  [ {{
"nuanced_content_to_be_learned":  [keyword style list of new and intellectually complex concepts
learned in this passage],
"passage":  "The passage text goes here."
}},
.... ],
"multiple_choice_question":  {{
"question":  "MC question utilizing the complex ideas learned in the passages.",
"options":  ["Option 1", "Option 2", "Option 3", "Option 4"] (do not use any indexing),
"answer_label":  "The correct answer label.  Return the exact text from options"
"step_by_step_answer_explanation":  "a detailed step-by-step layout of how one arrives at this answer
and what relevant information from the passages led to this answer."
}}
}}
```

**_Multi-Topic Textbook Academic Persona_ Prompt Template**

```
# Task
Generate consecutive passages in an academic textbook style, utilizing the following instructions.

## Instructions
- Write an extensive and detailed course unit suitable for a textbook targeted at specified persona.
You will be given a list of persona and need to select the most suitable one for the content
generation.
- You will be given a list of topics and subtopics for each topic. You need combine the suitable
topics and subtopics for the content generation. If there is no suitable combination, just use one
topic and all of its subtopics.
- Assume the reader already has a basic knowledge of the high-level topic, but they are looking to
learn more about subtopics.
- Engagement: Write with an academic, professional and engaging tone that captivates interest.
- Application: Incorporate specific, practical examples, such as proofs in calculus or critical
dates and figures in history.
- Generate 3-5 consecutive passages exploring the subject, increasing in nuance and detail by
passage, by that, I mean, increase the detail and example use of what the reader might learn from
the text.
- For each passage, you can select from the list of relevant keywords to guide the content of the
passages.
- Lastly, follow up the passages with a multiple choice question to test the most complex ideas in
learned from the passages, this will serve as a tool for the reader to test what they have learned
from this textbook.
Do not include a title or an introduction, simply write the content without headlines and
introductory phrases. Do not use images.

## Topic
{topic}

## Subtopic
{subtopic}

## Keyword
{keyword}

## Persona
{persona}

## Output
- Your output must be in the following JSON format:
{{ "passages": [ {{
"nuanced_content_to_be_learned": [keyword style list of new and intellectually complex concepts
learned in this passage],
"passage": "The passage text goes here."
}},
.... ],
"multiple_choice_question": {{
"question": "MC question utilizing the complex ideas learned in the passages.",
"options": ["Option 1", "Option 2", "Option 3", "Option 4"] (do not use any indexing),
"answer_label": "The correct answer label. Return the exact text from options"
"step_by_step_answer_explanation": "a detailed step-by-step layout of how one arrives at this answer
and what relevant information from the passages led to this answer."
}}
}}
```

**_Multi-Topic Blogpost Persona_ Prompt Template**

```
# Task
Generate consecutive passages in a blog post style, utilizing the following instructions.
## Instructions
- Write an informative and insightful blog post targeted at specified persona.  You will be given a
list of persona and need to select the most suitable one for the content generation.
- You will be given a list of topics and subtopics for each topic.  You need combine the suitable
topics and subtopics for the content generation.  If there is no suitable combination, just use one
topic and all of its subtopics.
- Assume the reader already has a basic knowledge of the high-level topic, but they are looking to
learn more about subtopics.
- Generate 3-5 consecutive passages exploring the subject, increasing in nuance and detail by
passage, by that, I mean, increase the detail and example use of what the reader might learn from
the text.
- For each passage, you can select from the list of relevant keywords to guide the content of the
passages.
- Your post should delve into the nuances of the topic, offering fresh perspectives and deeper
analysis.
- Inform:  Provide valuable, well-researched information that educates the reader.
- Engage:  Write in a conversational tone that connects with the audience, making complex ideas
accessible.
- Illustrate:  Use examples, anecdotes, or personal experiences to bring the topic to life.
- Lastly, follow up the passages with a multiple choice question to test the most complex concepts in
learned from the passages, this will serve as a tool for the reader to test what they have learned
from this blog post.  Do not give a title and do not start with sentences like "Have you ever..." or
"Hello dear readers..", simply write the content without these introductory phrases.

## Topic
{topic}

## Subtopic
{subtopic}

## Keyword
{keyword}

## Persona
{persona}

## Output
- Your output must be in the following JSON format:
{{ "passages":  [ {{
"nuanced_content_to_be_learned":  [keyword style list of new and intellectually complex concepts
learned in this passage],
"passage":  "The passage text goes here."
}},
.... ],
"multiple_choice_question":  {{
"question":  "MC question utilizing the complex ideas learned in the passages.",
"options":  ["Option 1", "Option 2", "Option 3", "Option 4"] (do not use any indexing),
"answer_label":  "The correct answer label.  Return the exact text from options"
"step_by_step_answer_explanation":  "a detailed step-by-step layout of how one arrives at this answer
and what relevant information from the passages led to this answer."
}}
}}
```

## *Multi-Topic Wikihow Persona* Prompt Template

```
# Task
Generate consecutive passages in a Wikihow style, utilizing the following instructions.
## Instructions
- Write a long and very detailed tutorial that could be part of WikiHow targeted at specified
persona.  You will be given a list of persona and need to select the most suitable one for the
content generation.
- You will be given a list of topics and subtopics for each topic.  You need combine the suitable
topics and subtopics for the content generation.  If there is no suitable combination, just use one
topic and all of its subtopics.
- Assume the reader already has a basic knowledge of the high-level topic, but they are looking to
learn more about subtopics.
- Generate 3-5 consecutive passages exploring the subject, increasing in nuance and detail by
passage, by that, I mean, increase the detail and example use of what the reader might learn from
the text.
- For each passage, you can select from the list of relevant keywords to guide the content of the
passages.
- Include in depth explanations for each step and how it helps achieve the desired outcome, inluding
key tips and guidelines.
- Ensure clarity and practicality, allowing readers to easily follow and apply the instructions.  Do
not use images.,
- Lastly, follow up the passages with a multiple choice question to test the most complex concepts in
learned from the passages, this will serve as a tool for the reader to test what they have learned
from this WikiHow.  Do not include a title or an introduction, simply write the content without
headlines and introductory phrases.

## Topic
{topic}

## Subtopic
{subtopic}

## Keyword
{keyword}

## Persona
{persona}

## Output
- Your output must be in the following JSON format:
{{ "passages":  [ {{
"nuanced_content_to_be_learned":  [keyword style list of new and intellectually complex concepts
learned in this passage],
"passage":  "The passage text goes here."
}},
.... ],
"multiple_choice_question":  {{
"question":  "MC question utilizing the complex ideas learned in the passages.",
"options":  ["Option 1", "Option 2", "Option 3", "Option 4"] (do not use any indexing),
"answer_label":  "The correct answer label.  Return the exact text from options"
"step_by_step_answer_explanation":  "a detailed step-by-step layout of how one arrives at this answer
and what relevant information from the passages led to this answer."
}}
}}
```

## F.2   GPT-4O GENERATION OUTPUT EXAMPLES

### F.2.1   *Topic*

---

**_Topic_ Prompt Output**

In the realm of Computing Education, particularly within the context of curriculum design, CS1 and CS2 are crucial introductory courses in computer science.  Here, CS1 typically serves as an entry point to programming, emphasizing foundational concepts such as variables, control structures, and basic algorithms.  Educators often focus on instilling strong problem-solving skills and a deep understanding of the software development lifecycle.  This stage is where students typically encounter key elements like pseudocode and flowcharting, which help them to plan and visualize their code before actually writing it.  Building upon the groundwork laid in CS1, the CS2 curriculum delves deeper into intermediate programming topics and advanced problem-solving techniques.  This course emphasizes the use of data structures like arrays, linked lists, stacks, queues, and trees. Students learn how to analyze the efficiency of algorithms using Big O notation and gain a more sophisticated understanding of object-oriented programming (OOP) principles.  Mastery of principles such as encapsulation, inheritance, and polymorphism becomes central as students work on more complex projects and begin to appreciate the design and implementation of more advanced software systems. One significant aspect that becomes apparent in CS2 is the utilization of Application Programming Interfaces (APIs).  APIs, such as API 817, 721, 628, 556, and 759, are critical for enabling communication between different software components.  In advanced CS2 courses, students learn to integrate APIs into their projects to extend functionality without reinventing the wheel.  This includes understanding how to read API documentation, making appropriate API calls, and implementing error handling effectively.  Additionally, students focus on integrating various software components and the importance of rigorous testing methodologies to ensure their software operates reliably and efficiently.
Can you answer this?
Which of the following best describes the significance of learning API utilization in a CS2 course?
1.  It enhances the students' ability to extend software functionality and ensures efficient communication between software components.
2.  It helps students to improve their understanding of variables and control structures.
3.  It offers a deeper understanding of data structures and basic algorithms.
4.  It introduces students to the basic software development lifecycle and rudimentary programming concepts.
Key:  It enhances the students' ability to extend software functionality and ensures efficient communication between software components.
The correct option is chosen based on the passage that explains how learning API utilization in a CS2 course helps students integrate and extend software functionality.  APIs are crucial for enabling communication between different software components, as described in the third passage.  This choice stands out because the other options refer to concepts primarily covered in introductory stages (CS1) or different aspects of CS2, such as data structures and basic programming constructs.

---

2484
2485
2486
2487
2488
2489
2490
2491
2492
2493
2494
2495
2496
2497
2498
2499
2500
2501
2502
2503
2504
2505
2506
2507
2508
2509
2510
2511
2512
2513
2514
2515
2516
2517
2518
2519
2520
2521
2522
2523
2524
2525
2526
2527
2528
2529
2530
2531
2532
2533
2534
2535
2536
2537

### F.2.2 *Topic Styles*

---

**_Topic Textbook Academic_ Prompt Output**

In the landscape of modern engineering, the materials we use often define the boundaries of what we can achieve. Fiber-reinforced composites, with their ability to cater to specific engineering needs, have transformed numerous industries. The differentiation between fiber-reinforced and particulate composites is rooted in their structure. Fiber-reinforced composites, as the name suggests, rely on fibers|often made of glass, carbon, or aramid|that provide significant strength and stiffness. These fibers are embedded within a matrix material, typically a polymer resin, which binds them together and distributes loads among them.

Think of fiber-reinforced composites as a team of collaborating specialists. The fibers, with their high tensile strength, handle most of the stresses, while the matrix supports the fibers and helps maintain their spatial arrangement. This results in a marriage of properties, where the composite exhibits a behavior that individual components could not achieve alone. By contrast, particulate composites utilize small particles dispersed throughout the matrix. Each particle contributes to the composite's overall performance, but lacks the directional strength that fiber reinforcements provide. Learning to understand these distinctions and their implications allows engineers to select the appropriate composite for the task at hand. The stress-strain behavior of fiber-reinforced composites offers a fascinating display of mechanical symbiosis. When subjected to external loads, the fiber within the composite seeks to elongate based on its own high tensile capabilities. However, the matrix material, being more ductile and less stiff, stretches and distributes this load across the fiber network. This intricate dance between the matrix and the fibers is what defines the unique mechanical properties of fiber-reinforced composites.

Consider the mechanics of load transfer: when a force is applied, it is not merely the fibers or the matrix acting independently. For instance, if an external tension is applied, fibers will deform elastically and the matrix will distribute this load, transferring stress efficiently throughout the composite. The efficiency of this load transfer depends not only on the intrinsic properties of the fibers and the matrix but also on the quality of the bond at their interface. A strong interfacial bond ensures that the stress transfer is seamless, mitigating potential failure points. Recent studies utilizing techniques have examined the durability and efficiency of these bonds under various stress conditions, shedding light on failure mechanisms and enhancing material reliability.

Exploring advanced manufacturing techniques reveals the meticulous craftsmanship behind creating high-performance fiber-reinforced composites. Methods such as filament winding, pultrusion, and resin transfer molding (RTM) are designed to maximize the alignment and distribution of fibers, ensuring the composite's robust mechanical properties. Filament winding, for example, involves winding continuous fibers soaked in resin around a mandrel, which is then cured to form a solid shape. This technique is particularly advantageous in producing high-pressure vessels and piping, where the controlled fiber alignment translates to superior strength. Recent guidelines have been instrumental in standardizing these manufacturing processes, ensuring consistency and reliability across applications.

Moreover, interface behavior and bonding strength are critical factors in determining the performance of fiber-reinforced composites. High-quality bonding at the fiber-matrix interface can significantly enhance the composite's durability. The strength of this interface can be quantified through various testing methodologies, including those highlighted standards. By subjecting samples to mechanical stresses and examining failure modes, engineers can better understand and improve the bonding strategies. These insights are pivotal in applications where material failure can have catastrophic consequences, such as in aerospace and structural engineering.

Q: Which advanced technique is particularly advantageous for producing high-pressure vessels and relies on continuous fiber alignment?
i.  Resin transfer molding
ii.  Filament winding
iii.  Pultrusion
iv.  Particulate dispersion
Key:  ii
Filament winding is a technique where continuous fibers soaked in resin are wound around a mandrel and then cured to form a solid shape. This method ensures controlled fiber alignment, making it particularly advantageous for producing high-pressure vessels and piping, as described in the text. Particulate dispersion is irrelevant for fiber-reinforced composites, while resin transfer molding and pultrusion, though useful, are not mentioned as the go-to method for such applications.

2538
2539
2540
2541
2542
2543
2544
2545
2546
2547
2548
2549
2550
2551
2552
2553
2554
2555
2556
2557
2558
2559
2560
2561
2562
2563
2564
2565
2566
2567
2568
2569
2570
2571
2572
2573
2574
2575
2576
2577
2578
2579
2580
2581
2582
2583
2584
2585
2586
2587
2588
2589
2590
2591

## *Topic Textbook Narrative* Prompt Output

The lithosphere, the outermost shell of the Earth, is divided into several large and small tectonic plates. These plates float on the semi-fluid asthenosphere below. The interactions at the edges of these plates, known as plate boundaries, define the primary processes of plate tectonics. Plate boundaries are classified into three main types: divergent, convergent, and transform. At divergent boundaries, plates move apart from each other. The mid-Atlantic Ridge is a prime example, where volcanic activity creates new oceanic crust. At convergent boundaries, plates move towards each other, leading to subduction or mountain building. The Andes mountain range illustrates this process where the oceanic plate subducts beneath the continental plate. Transform boundaries, such as the San Andreas Fault in California, are characterized by lateral sliding of plates against each other. Understanding the kinematics of fault zones is crucial in structural geology and tectonics. Fault zones exhibit complex behaviors depending on various factors, such as stress, temperature, and material properties. Mechanical behavior of fault zone materials can range from brittle fracturing to ductile flow. Finite strain analysis provides insights into the cumulative deformation experienced by materials in fault zones. For instance, by applying finite strain analysis to fault rocks in the exhumed sections of the Alpine Fault in New Zealand, researchers have revealed rotational kinematics and strain partitioning that occurred over geological timescales. These analyses help in understanding the evolution of faults and their current dynamics. Geodynamic processes encompass the large-scale movements and deformation within the Earth's interior that drive plate tectonics and fault dynamics. Thermo-mechanical modeling is an essential tool in geodynamics, helping to understand how heat affects tectonic processes. By simulating temperature and mechanical conditions within the Earth's crust and mantle, scientists can predict the behavior of tectonic plates and the development of fault zones. For instance, thermo-mechanical models have shown that increased mantle heat flow beneath the East African Rift contributes to the thinning lithosphere and the formation of rift valleys. This model illuminates the significant role of temperature in influencing the mechanical properties and tectonic evolution of plate boundaries.
Which process contributes to the formation of rift valleys according to thermo-mechanical models??
i.  Increased mantle heat flow
ii.  Mountain building at convergent boundaries
iii.  Lateral sliding at transform boundaries
iv.  Subduction of oceanic crust
A: i
Thermo-mechanical models are used to understand the role of heat in tectonic processes. According to these models, increased mantle heat flow beneath the lithosphere contributes to its thinning and the formation of rift valleys. This process can be particularly observed in regions like the East African Rift, where thermal effects play a crucial role in tectonic evolution.

## *Topic Blogpost* Prompt Output

Phylogenetics and molecular evolution form core concepts in understanding the evolutionary relationships among species. In computational ecology, these methods are critical for constructing phylogenetic trees using algorithms that can process vast genomic datasets. Starting with the fundamentals, phylogenetics involves the study of the evolutionary history and relationships among individuals or groups of species through DNA sequencing data. Molecular evolution focuses on evolutionary changes at the molecular level, particularly within DNA, RNA, and protein sequences. Basic tools for these analyses include software like MEGA and BEAST, which facilitate phylogenetic tree construction, and databases like GenBank that provide access to nucleotide sequences. Building on these fundamentals, advanced phylogenetic techniques integrate more intricate aspects like hybridization and introgression, which can offer deeper insights into species evolution. Hybridization occurs when two different species or subspecies interbreed. Introgression refers to the incorporation of genes from one species into the gene pool of another through repeated backcrossing. These processes can generate novel genetic combinations that may enhance adaptability and lead to new species formations. Computational tools such as STRUCTURE and fastSTRUCTURE are used to analyze complex patterns of population structure and hybridization, revealing introgression events and tracing evolutionary histories with greater precision. For a more concrete understanding, let's delve into specific case studies where hybridization and introgression have played a significant role. For example, the Heliconius butterflies exhibit extensive hybridization among populations, enabling researchers to study adaptive radiation|a process through which organisms diversify rapidly into a multitude of new forms. Computational tools have allowed scientists to trace gene flow amongst these butterflies by comparing genomic data from different populations. Another illustrative example involves the hybridization between polar bears and brown bears. The genetic introgression resulting from their interaction has highlighted how climate change affects species adaptation and evolution. These studies underscore the importance of high-resolution genomic data and sophisticated algorithms in uncovering the nuanced evolutionary dynamics impacted by hybridization and gene flow.
Can you answer this?
Which of the following best describes the role of introgression in evolutionary biology?
(1) Introgression is the linear transfer of genetic traits from parent species without any backcrossing.
(2) Introgression occurs solely within the same species and involves no external gene flow.
(3) Introgression involves the transfer of genes from one species into the gene pool of another through hybridization and repeated backcrossing.
(4) Introgression is the process by which entirely new species are formed without any genetic information exchange.
A: Introgression involves the transfer of genes from one species into the gene pool of another through hybridization and repeated backcrossing.
The correct answer is derived from understanding the detailed description of introgression given in the passages. The passages explain that introgression involves hybridization and repeated backcrossing, which facilitates the incorporation of genes from one species into the gene pool of another. This process is characterized by the movement of genetic material across species boundaries, contributing to evolutionary dynamics and species adaptation.

---

*Topic Wikihow* Prompt Output

Histoplasmosis is a fungal infection caused primarily by Histoplasma capsulatum, a dimorphic fungus. The initial diagnosis often begins with a thorough patient history and physical examination. Clinicians should particularly inquire about any travel histories, as Histoplasma capsulatum is commonly found in specific endemic regions such as the Ohio and Mississippi River valleys in the United States. Clinically, histoplasmosis presents with a range of symptoms from mild flu-like illness to severe disseminated disease, especially in immunocompromised individuals. Key signs include fever, cough, and lymphadenopathy. Epidemiologically, understanding the patient's environmental exposure is critical for diagnosis, and knowing the typical endemic regions can help clinicians consider histoplasmosis as a differential diagnosis early on. Once clinical suspicion is established, laboratory diagnostic techniques are crucial for confirming histoplasmosis. One of the primary methods is the culture of bodily fluids such as blood, sputum, or tissue biopsies, which can take several weeks due to the slow-growing nature of Histoplasma capsulatum. For microscopic examination, samples can be stained using techniques like Giemsa or Wright's stain, which allow visualization of the fungal elements within macrophages where Histoplasma resides intracellularly. Histoplasma can appear as small, oval yeasts measuring 2-4 micrometers in diameter. Due to the lengthy culture process, direct microscopic examination and fungal stains are valuable for rapid provisional diagnosis while awaiting culture results. In addition to cultures and microscopic examination, several serological tests are available for diagnosing histoplasmosis. Tests such as the complement fixation test and immunodiffusion can detect antibodies against Histoplasma antigens. These serological tests are useful but can have limitations, such as cross-reactivity with other fungal pathogens. Histoplasma antigen detection, typically from urine or blood samples, offers another useful diagnostic method. The presence of Histoplasma polysaccharide antigen can be a strong indicator of active infection, particularly in disseminated disease. Molecular diagnostic tools like polymerase chain reaction (PCR) amplify specific Histoplasma DNA, providing rapid and highly specific identification. These modern methods facilitate early diagnosis and intervention, which is crucial for patient outcomes, especially in severe cases.
Test your knowledge:
In a patient suspected of Histoplasmosis based on travel history to the Ohio River valley and presenting with fever, cough, and lymphadenopathy, what is the most rapid initial diagnostic test?
* Complement fixation serological test
* Histoplasma culture from blood
* Histoplasma antigen detection in urine
* Giemsa-stained microscopic examination of sputum
Answer: Giemsa-stained microscopic examination of sputum
The Giemsa-stained microscopic examination of sputum is the most rapid initial diagnostic test among the options provided. Histoplasma culture from blood, although definitive, takes several weeks. Complement fixation serological tests are useful but can take longer to process and interpret. Histoplasma antigen detection in urine is also rapid and useful but is not mentioned as the most rapid compared to direct microscopic examination. Hence, Giemsa-stained microscopic examination, which allows visualization of the fungi within hours, is the fastest diagnostic test in this context.

---

### F.2.3 *Topic Styles Persona*

---

*Topic Textbook Academic Persona* Prompt Output

One might think of the immune system as a highly disciplined army, well-prepared to fend off any invaders. But, behind this regiment lies a covert operative unit that guides the immune army's strategies: epigenetic modifications. Recent advances like next-generation sequencing (NGS) have revolutionized our understanding of these hidden layers of gene regulation. NGS allows scientists to scrutinize DNA and RNA to pinpoint epigenetic markers|modifications that don't change the DNA sequence but still direct how genes are expressed. These markers orchestrate the opening and closing of chromatin structures, like gates controlling access to vital installation parts of the immune cells' operational code. Fundamentally, this defines how immune responses are magnified or subdued in various contexts. Dive deeper, and you'll find that immune cells don't just stumble upon epigenetic and metabolic paths; they are finely tuned by an intricate signaling symphony. The mTOR pathway, a central regulator of cell metabolism and proliferation, plays maestro here. This pathway integrates signals from nutrients, energy status, and growth factors, impacting cellular metabolism and immunity. When immune cells activate, there's a substantial increase in glucose metabolism|a swift switch to glycolysis|ensuring that these cells have the rapid energy needed for their defensive roles. This metabolic reprogramming is akin to mobilizing an elite task force with the best rations and supplies, empowering them to endure and excel in a short but intense operation. Imagine now a scenario where this immune response goes haywire. Reactive oxygen species (ROS) generated during these metabolic processes can cause collateral damage, leading to chronic inflammation or autoimmune diseases. Scientists have recently started exploring mTOR pathway inhibitors as a form of intervention, essentially applying the brakes on this runaway train. By fine-tuning this metabolic conduit, researchers hope to harness the precision of targeted epigenetic modifications and therapeutic interventions. This approach might prevent the overzealous immune responses by restoring metabolic equilibrium and precise gene regulation. It's like having a master planner who not only directs the troops but also ensures the health of their operational environment, reducing the likelihood of destructive friendly fire.
Can you answer this?
Which therapeutic approach precisely targets the metabolic and epigenetic pathways to manage overactive immune responses?
A. Glucose metabolism enhancement
B. Next-generation sequencing
C. mTOR pathway inhibitors
D. Reactive oxygen species management
Selected answer: mTOR pathway inhibitors

---

*Topic Textbook Narrative Persona* Prompt Output

The Momentum Contrast (MoCo) framework is a widely used method for unsupervised learning in neural networks. At its core, MoCo aims to map different augmentations of the same image to nearby points in the feature space, thereby learning useful feature representations. One key component of MoCo is the Queue mechanism, which maintains a large set of negative samples. This Queue allows the model to update efficiently, even as new data is introduced, without recalculating the entire set of negative samples. For instance, in a neural network designed to classify agricultural images, the Queue would store various negative samples of plant diseases that do not match the positive sample, thereby refining the model's ability to differentiate and classify correctly. Building upon the basic MoCo framework, several variants like MoCo-k have been introduced to enhance model performance. MoCo-k variants, for example, optimize the size and dynamic updating of the Queue, providing better utilization of hardware resources. Layer Normalization is another critical component that ensures each layer within the neural network maintains a standardized range of output values, improving training stability and speed. When combined with contrastive learning techniques, these enhancements significantly advance the capacity of neural networks to learn from unlabeled data. For example, in detecting subtle variations in hybrid plant varieties, MoCo-k can streamline the process, ensuring the network focuses on the essential features of the images. The MoCo-c framework introduces further refinements tailored for large-scale data environments. MoCo-c focuses on dynamically adjusting the Queue's composition and size based on the training phase, thereby optimizing training efficiency. This dynamic Queue management enables the model to handle diverse and large-scale datasets more effectively. Practical applications, such as identifying the optimal traits in hybrid plant varieties for commercial agriculture, benefit from MoCo-c by reducing the computational burden and accelerating the learning process. This optimization ensures that the model adapts quickly to new data patterns, providing more accurate and timely predictions.
Try to solve this:
Which component of the MoCo framework is primarily responsible for maintaining a set of negative samples that update efficiently as new data is introduced?
1) MoCo-k
2) Layer Normalization
3) Queue mechanism
4) MoCo-c
Answer: Queue mechanism
The passages explain that the Queue mechanism is a key component of the MoCo framework, specifically designed to maintain and update a large set of negative samples efficiently. As new data is introduced, this mechanism ensures that the network does not need to recalculate the entire set of negative samples, thus optimizing training efficiency. While Layer Normalization and variants like MoCo-c and MoCo-k contribute to other aspects of neural network optimization, the primary function of handling negative samples belongs to the Queue mechanism.

*Topic Blogpost Persona* Prompt Output

In the realm of hydrogeology, understanding groundwater flow is pivotal, particularly in the context of matrix flow. Unlike macroscopic or preferential flow through cracks and fissures, matrix flow refers to the movement of groundwater through the porous spaces within a geological matrix. This type of flow is significantly influenced by the hydraulic conductivity of the material, which is a measure of a substrate's ability to allow water to pass through it. For instance, sand and gravel vastly differ in their hydraulic conductivity compared to clay. By exploring these differences, we can predict how quickly groundwater can rejuvenate aquifers or how contaminants might spread. Thus, the comprehension of matrix flow and hydraulic conductivity forms a foundational aspect of effective water resource management and pollution control. As we dig deeper into hydrology, it's essential to understand the concept of baseflow, which is the portion of streamflow that comes from groundwater seeping into rivers. This stable flow maintains river levels during dry periods and impacts the overall health of aquatic ecosystems. Now, to appreciate the influence of various factors on baseflow, we can utilize isohyets. Isohyets are contours that connect points of equal rainfall on a map. By analyzing isohyets, hydrologists can interpret how rainfall distribution impacts groundwater recharge and subsequently, baseflow levels. For example, areas with dense isohyets indicating high rainfall typically have higher groundwater recharge rates, contributing more to baseflow over time. Well logging is a crucial methodology used to acquire detailed records of geological formations penetrated by boreholes. This technique involves the measurement of physical properties through well logs to characterize subsurface conditions effectively. There are various types of well logs such as electrical, sonic, and nuclear, each offering unique insights into the geological and hydrological conditions. Electrical logging can indicate the presence of water or hydrocarbons within aquifers, while sonic logging provides data on rock hardness and porosity, further aiding in the evaluation of hydraulic conductivity. Therefore, well logging serves as an indispensable tool for groundwater studies, fostering better-informed decisions in water management and environmental protection.
Try to solve this:
Which technique would you use to determine the hydraulic conductivity of subsurface layers in a borehole?
1. Baseflow
2. Isohyets
3. Well logging
4. Matrix flow
Answer: 3
To determine the hydraulic conductivity of subsurface layers, you need detailed information about the geological formations penetrated by a borehole. 'Well logging' is specifically used to obtain this detailed data through various logs such as electrical, sonic, and nuclear, helping in the evaluation of hydraulic conductivity. Matrix flow, baseflow, and isohyets do not provide such detailed subsurface information and are more focused on groundwater flow characteristics and rainfall patterns.

**2700**
**2701**

---

*Topic Wikihow Persona* Prompt Output

---

**2702** Incorporating Quest-Based Learning (QBL) into your classroom can significantly enhance student
**2703** engagement and drive learning outcomes. QBL involves structuring lessons so students complete
quests|individual or group tasks that mimic real-world challenges. For example, you might create
**2704** a quest where students need to solve a set of math problems to 'unlock' the next stage of their
project. Gamification Strategies like this help to foster a sense of achievement and motivation by
**2705** turning conventional assignments into interactive and immersive experiences. Alternate Reality Games
**2706** (ARGs) are an advanced form of QBL where students solve complex, multi-step problems. ARGs often
involve a narrative layer, making them feel like participants in a story rather than mere students
**2707** completing assignments. Integrating these elements requires thoughtful planning to ensure they
align with your learning objectives. One effective Gamification Strategy is Avatar Customization,
**2708** where students create and personalize their virtual representations within the learning environment.
Allowing students to customize their avatars can increase emotional investment in the tasks at hand.
**2709** This sense of ownership can translate to higher engagement levels and a deeper connection to the
material. Application Integration is crucial for seamless execution of these strategies. Tools
**2710** like Learning Management Systems (LMS) or specialized educational software can be integrated to
facilitate these customized learning experiences. For example, using an LMS, you can manage quests,
**2711** track progress, and provide feedback all within a unified platform. These tools often come with
**2712** plug-ins for various gamified features, making it easier to implement a student-centered learning
approach. Badge Systems are another vital component of gamification in education. Badges act as
**2713** visual representations of achievements, marking milestones like quest completions, top scores, or
peer collaboration. They serve as both intrinsic and extrinsic motivators|students feel internally
**2714** rewarded by their achievements while also perceiving external recognition. Implementing a Badge
**2715** System can encourage healthy competition and goal-setting among students. Additionally, these
systems enable Data-Driven Decision Making in the classroom. By analyzing which tasks earn the most
**2716** badges or noticing trends in student performance, educators can identify areas requiring additional
focus or support. This data can guide instructional strategies and personalize learning paths to
**2717** better meet each student's needs.
**2718** Q: Which aspect of Gamification Strategies can enhance emotional investment and engagement in student
tasks? Alternate Reality Games
**2719** Badge Systems
Avatar Customization
**2720** Quest-Based Learning
The answer is: Avatar Customization
**2721** Avatar Customization allows students to create and personalize their virtual representations, leading
to higher emotional investment in tasks. This personalization increases their engagement levels
**2722** and connection to the material, as highlighted in the second passage. Neither Quest-Based Learning
**2723** nor Alternate Reality Games uniquely offer this personalization aspect. Badge Systems primarily
focus on motivation through achievements and milestones rather than emotional investment through
**2724** customization.

**2725**
**2726**
**2727**
**2728**
**2729**

**2730** ## F.2.4    *Multi-Topic Styles Persona*

**2731**
**2732**

---

*Multi-Topic Textbook Academic Persona* Prompt Output

---

**2733**
**2734**

**2735** When considering the emotional impact of narratives, one can observe how thematic resonance
profoundly influences a reader's or viewer's engagement. Thematic resonance, in essence, refers
**2736** to the alignment of the narrative's central themes with the personal experiences and beliefs of the
audience. For example, a story focused on resilience in the face of adversity can evoke powerful
**2737** emotional responses from an audience that has experienced similar struggles. This resonance not
**2738** only enhances the emotional depth of the narrative but also ensures that the story remains memorable
and impactful. Interactive storytelling introduces a dynamic element to narrative construction,
**2739** where the player's choices and actions directly influence the unfolding plot. This mechanism of
player agency grants the player a sense of ownership over the story. For instance, in video games
**2740** like 'The Witcher 3' or 'Detroit: Become Human,' players make decisions that can lead to multiple
endings, thus personalizing the narrative experience. This interactivity fosters a deeper emotional
**2741** connection as players become more invested in the outcomes influenced by their decisions. Delving
deeper into interactive narratives, the concept of emotional branching becomes paramount. Emotional
**2742** branching refers to the development of different narrative paths based on the emotional choices
**2743** characters make. A choice made out of compassion might lead to a significantly different storyline
compared to one driven by revenge. These branches create psychological investments for players,
**2744** prompting them to consider the emotional weight of their decisions. The feedback loops generated
from these emotional choices and their subsequent narrative outcomes further enrich the storytelling
**2745** experience, making it a complex web of cause and effect that mirrors real-life decision-making
**2746** complexities.
Try to solve this:
**2747** Which concept best explains the phenomenon where players feel more connected to a game's story due to
their ability to make impactful decisions?
**2748** A. Emotional branching
**2749** B. Thematic resonance
C. Player agency
**2750** D. Narrative feedback loops
Selected answer: Player agency
**2751** The passages discussed various narrative dynamics, and the term 'player agency' specifically refers
to the mechanism by which players' decisions influence the storytelling, thereby creating a deeper
**2752** emotional connection. By allowing players to shape the story, they feel a greater sense of ownership
**2753** and engagement, making 'Player agency' the most appropriate answer.

---

**_Multi-Topic Textbook Narrative Persona_ Prompt Output**

Monetary and fiscal policies are vital components of a nation's economic framework. Understanding these concepts requires a deep dive into their designs and roles. Monetary policy, managed by a country's central bank, focuses on regulating the money supply and controlling inflation by adjusting interest rates. On the other hand, fiscal policy, orchestrated by the government, involves taxation and public spending to influence the economy. The design of taxation is critical; it must achieve a balance between encouraging economic growth and ensuring equitable distribution of wealth. Public spending, too, needs to be meticulously planned to maximize societal benefits while avoiding excessive debt. Taxation strategies vary widely and can significantly affect economic and social landscapes. Progressive taxation, where the tax rate increases as income increases, is designed to reduce inequality by placing a larger burden on those who can afford to pay more. In contrast, regressive taxation places a higher relative burden on lower-income individuals, often seen in sales taxes. Determining the priorities for public spending is equally intricate. Governments need to allocate funds efficiently across sectors like healthcare, education, and infrastructure while ensuring that expenditures do not outpace revenues, thus averting unsustainable debt growth. The impact of taxation on economic behavior is profound. High taxes can discourage investment and savings, while certain tax incentives can stimulate specific industries. Public spending on goods and services such as roads, schools, and hospitals enhances economic productivity and social well-being. Fiscal policy tools|such as subsidies, grants, and public investments|are employed to stabilize the economy. An essential consideration in the design of both tax policy and public spending is intergenerational equity; ensuring that current actions do not unfairly burden future generations. This requires a balance between meeting today's needs and planning for future sustainability.
Here is a question for you:
Which of the following best describes the concept of intergenerational equity in the context of fiscal policy?
I. Promoting regressive taxation to equalize economic opportunities.
II. Ensuring tax rates remain the same for all generations.
III. Balancing between meeting current needs and planning for future sustainability.
IV. Allocating public spending based on the immediate needs of the current population only.
The answer is: III
Intergenerational equity refers to the fair treatment of different generations, ensuring that current policies do not place undue burden on future generations. This involves a careful balance in fiscal policy between addressing the immediate needs (public spending) and planning for sustainability (taxation and public investment) for the future. The passages discussed how this balance is crucial to avoid excessive debt and ensure long-term economic stability.

**_Multi-Topic Blogpost Persona_ Prompt Output**

The adoption of genetically modified (GM) crops has become a pivotal point in modern agricultural biotechnology. With these advancements, farmers can achieve higher yields and more robust resistance to pests and diseases. However, technology adoption in the context of GM crops involves intricate biosafety regulations to ensure they are safe for both the environment and human consumption. Regulatory bodies such as the European Food Safety Authority play a crucial role in setting and enforcing these standards, ensuring that GM crops undergo rigorous assessment before they are approved for cultivation and sale. By maintaining high biosafety standards, these bodies aim to mitigate potential risks associated with the widespread use of such biotechnology. Beyond the technical and regulatory aspects, the socio-economic and ethical dimensions of GM crops also carry significant weight. For instance, Golden Rice, a genetically modified variety developed to combat vitamin A deficiency, exemplifies the potential public health benefits of GM crops. Despite its promise, the adoption of Golden Rice has faced socio-economic challenges, such as market acceptance and intellectual property issues. Seed patents, for example, often prevent farmers in developing countries from freely using and replanting GM seeds. Thus, while GM crops have the potential to contribute to food security and nutritional improvements, their integration into agricultural markets must navigate a complex landscape of ethical considerations and economic barriers. The commercialization and adoption of GM crops also follow a recognizable technology adoption curve, which categorizes adopters into innovators, early adopters, early majority, late majority, and laggards. Seed patents play a critical role in this process, often protecting the intellectual property of the companies that develop these technologies. However, they can also create barriers for widespread adoption, particularly in less developed agricultural markets. For instance, small-scale farmers might struggle with the higher costs associated with patented seeds, limiting their ability to benefit from agricultural biotechnology. Therefore, addressing these disparities is crucial for optimizing the benefits of GM crops across various markets.
Challenge: Which regulatory body plays a crucial role in the adoption of genetically modified crops by ensuring biosafety standards?
1) United Nations
2) European Food Safety Authority
3) World Trade Organization
4) International Seed Federation
Answer: European Food Safety Authority
The first passage elaborates on the role of the European Food Safety Authority in setting and enforcing biosafety standards for GM crops. This body ensures that these crops undergo rigorous assessments before approval. None of the other options are mentioned in the passages as having similar roles in biosafety regulation for GM crops.

---

**_Multi-Topic Wikihow Persona_ Prompt Output**

Representing parties involved in doping allegations or investigations requires a deep understanding of the intricacies of sports anti-doping laws. National Anti-Doping Organizations (NADOs) play a critical role in maintaining the integrity of sports by conducting regular drug tests on athletes and managing the results. When an athlete's sample results in an Adverse Analytical Finding (AAF), it indicates the presence of a prohibited substance. However, not all cases of AAFs are straightforward. Tainted supplements are a common issue, where an athlete may unknowingly consume a prohibited substance due to contamination or mislabeling of dietary products. Legal representatives must be equipped with the knowledge to challenge AAFs by investigating the sources of supplements and proving any unintended consumption. In doping cases, understanding the standards of proof is paramount. Unlike criminal law, where the standard is 'beyond a reasonable doubt,' doping allegations are typically established on a 'balance of probabilities.' This means that the evidence must show that it is more likely than not that a doping violation occurred. Legal practitioners need to be adept at presenting and challenging evidence to meet or contest this standard. Anti-Doping Organizations (ADOs) are responsible for ensuring that the testing and results management processes are transparent and fair. When representing clients in hearings, it's crucial to scrutinize the proper adherence to procedural steps by ADOs. Legal representatives must be skilled in cross-examining witnesses, presenting counter-evidence, and leveraging expert testimonies to argue their case effectively. Therapeutic Use Exemptions (TUEs) are a critical aspect of the anti-doping landscape. Athletes with legitimate medical conditions can apply for TUEs to use prohibited substances that are necessary for their health. Legal counsel must be well-versed in the criteria and approval process for TUEs, as well as the documentation required to support an application. Additionally, understanding mitigating circumstances that may reduce the severity of sanctions is crucial. Factors such as the athlete's level of fault, cooperation with anti-doping authorities, and the specifics of the contamination or misuse situation can impact the outcomes. Familiarity with case law and legal precedents allows legal professionals to reference past decisions that may influence current cases. This comprehensive approach ensures that every potential avenue for defense is explored, providing their clients with the best possible representation.
What is the standard of proof typically required in doping allegations??
Preponderance of the evidence
Clear and convincing evidence
Beyond a reasonable doubt
Balance of probabilities
The correct choice is: balance of probabilities.
The standard of proof required in doping allegations is the 'balance of probabilities.' This means that the evidence must show that it is more likely than not that a doping violation occurred. This is different from the 'beyond a reasonable doubt' standard used in criminal cases. A balance of probabilities standard is a lower threshold, appropriate for civil and most administrative cases, including those governed by anti-doping regulations.

---

## F.3 GPT-3.5 GENERATION OUTPUT EXAMPLES

---

**_Topic Styles Persona_ Prompt Output**

Rydberg atoms are highly excited states of an atom that can have a principal quantum number much larger than the ground state. These atoms have been studied for their unique properties in the field of quantum information. Rydberg atoms have long coherence times and strong dipole-dipole interactions, which make them ideal for quantum memory and communication applications. In recent years, researchers have made significant progress in using Rydberg atoms to store and manipulate quantum information. By using laser pulses to excite the Rydberg atoms, researchers have demonstrated the ability to store quantum information in the form of superposition states. This has the potential to revolutionize the field of quantum computing. Quantum teleportation is a process by which quantum information can be transferred from one location to another without physically transmitting the information. This is made possible by the phenomenon of quantum entanglement, which is a strong correlation between two particles that can exist even when they are separated by large distances. In order to perform quantum teleportation, two particles must be entangled and a third particle, which contains the quantum information to be teleported, must be prepared in a special state called a Bell state. When the third particle is measured, the state of the other two particles is instantly determined, allowing the quantum information to be transferred. This process has been demonstrated experimentally using a variety of different systems, including photons and ions. Vacuum fluctuations are fluctuations in the electromagnetic field that exist even in the absence of any particles. These fluctuations have been studied extensively in the context of atom optics, which is the manipulation of atoms using light. One application of atom optics is the creation of photonic crystals, which are materials that have a periodic variation in their refractive index. These materials can be used to control the propagation of light and have potential applications in quantum information processing. By using vacuum fluctuations to manipulate the motion of atoms, researchers have been able to create photonic crystals with a high degree of control over their properties. This has the potential to lead to the development of new types of quantum devices that are based on the manipulation of light and matter.
What is the process by which quantum information can be transferred from one location to another without physically transmitting the information?
Quantum entanglement
Bell state
Vacuum fluctuations
Atom optics
Answer: Quantum entanglement
Quantum teleportation is made possible by the phenomenon of quantum entanglement, which is a strong correlation between two particles that can exist even when they are separated by large distances. When two particles are entangled and a third particle, which contains the quantum information to be teleported, is prepared in a special state called a Bell state, the state of the other two particles is instantly determined, allowing the quantum information to be transferred.

## F.4 Llama-3.1-Instruct-8B Generation Output Examples

**Topic Styles Persona Prompt Output**

As a senior citizen living alone and facing food insecurity, one of the most significant challenges is accessing transportation that meets your needs. Understanding modal split is crucial in this context, as it refers to the distribution of transportation modes used by individuals in a particular area. The modal split can be influenced by various factors, including the availability and accessibility of transportation infrastructure, such as bus stops, train stations, or bike lanes. For instance, a well-planned transportation system with efficient public transportation can encourage people to use buses or trains, reducing the reliance on private vehicles and promoting a more balanced modal split. The transportation network is a complex system that encompasses various transportation modes, including public transportation, private vehicles, and non-motorized modes such as walking and cycling. To ensure that the transportation network meets the needs of senior citizens like yourself, it is essential to consider the characteristics of each transportation mode. For example, public transportation often requires a certain level of physical mobility, whereas non-motorized modes can be more accessible for those with mobility issues. The availability and accessibility of transportation infrastructure, such as bus stops, train stations, or bike lanes, also play a significant role in determining the effectiveness of the transportation network. Transportation demand management (TDM) strategies can significantly impact the modal split and the overall effectiveness of the transportation network. TDM involves managing the demand for transportation by promoting the use of alternative modes, such as carpooling, ride-sharing, or public transportation. For instance, implementing smart traffic signals that adjust to real-time traffic conditions can encourage drivers to use alternative modes, reducing congestion and promoting a more balanced modal split. Additionally, TDM strategies can be used to manage the demand for transportation infrastructure, such as building more bike lanes or improving public transportation services.
Here is a question for you:
What is the primary goal of transportation demand management (TDM) strategies in the context of senior citizens' transportation needs?
* To reduce the reliance on private vehicles
* To increase the use of public transportation
* To promote carpooling and ride-sharing
* To improve the accessibility of transportation infrastructure
The correct choice is: reduce the reliance on private vehicles
To arrive at this answer, first, consider the context of senior citizens' transportation needs, which often involves accessing transportation that meets their specific requirements. Then, recall the definition of TDM strategies, which involves managing the demand for transportation to promote the use of alternative modes. Finally, relate the primary goal of TDM strategies to the context of senior citizens' transportation needs, and you will arrive at the conclusion that the primary goal of TDM strategies is to reduce the reliance on private vehicles, promoting a more balanced modal split and improving the overall effectiveness of the transportation network.

## F.5 Mistral-Instruct-7B Generation Output Examples

**Topic Styles Persona Prompt Output**

The Type D Personality, characterized by high levels of negative affectivity and social inhibition, is often associated with a heightened risk of various health problems. This personality type is more prone to emotional dysregulation, which refers to the inability to manage and regulate emotions effectively. In response to stress, Type D individuals tend to exhibit reactive coping behaviors, such as withdrawing socially or becoming overly emotional, which can further exacerbate their health issues. Contrary to the fight-or-flight response, the tend-and-befriend response is a more subtle strategy employed by some individuals in response to stress. This response involves seeking social support and comfort from others. However, for individuals with a Type D personality and emotional dysregulation, the tend-and-befriend response can be challenging to execute effectively, potentially leading to further emotional distress. The Response-Based Model of Stress proposes that individuals respond differently to stressors based on their personal and social resources. For Type D individuals, their tendency towards emotional dysregulation and reactive coping may be a result of limited social and personal resources, making it difficult for them to manage stress effectively. Environmental metaphysics and ontology delve into the nature of the environment and its relationship with human existence. Within this field, environmental ontology is a key concept, focusing on the essence and fundamental characteristics of the environment. It seeks to understand the environment as a dynamic and interconnected system, composed of various entities and processes, and how these components interact and affect each other. One essential aspect of environmental ontology is the study of human-environment interaction. This area examines the ways in which humans interact with and influence the environment, as well as how the environment in turn impacts human beings. By understanding these interactions, we can better appreciate the complex, reciprocal relationship between humans and their environment. In the study of environmental ontology, there are two primary viewpoints: anthropocentric and biocentric. The anthropocentric viewpoint holds that humans are the central and most important part of the environment, while the biocentric viewpoint argues that all living beings have equal value and should be considered in environmental discussions. By exploring these viewpoints, environmental ontology highlights the importance of ethical considerations in environmental debates and decision-making.
Can you answer this?
Which of the following is a strategy exhibited by a Type D personality in response to stress?
* Fight
* Flight
* Tend
* Befriend
Key: Tend
A Type D personality tends to seek social support and comfort from others in response to stress, which is the tend-and-befriend response. This is different from fight or flight responses, which involve either confronting or avoiding the stressor.

