# OpenReview forum: "On the Diversity of Synthetic Data and its Impact on Training Large Language Models"
_ICLR.cc/2026/Conference — Submitted to ICLR 2026_

### Official Review · Reviewer_64PC · 2025-10-27

**Soundness:** 2
**Presentation:** 1
**Contribution:** 2
**Rating:** 4
**Confidence:** 4

**Summary:**

This paper introduces a novel metric, the "LLM cluster-agent," designed to quantify the diversity of synthetic datasets. The proposed pipeline leverages a large language model (LLM) to iteratively generate clustering criteria (metadata and metrics) and subsequently group data samples, with the resulting "LLM cluster score" serving as a proxy for diversity. The authors conduct a series of controlled experiments, training 350M and 1.4B parameter models, to investigate the correlation between this diversity score and downstream model performance (in both pre-training and supervised fine-tuning settings). The study manipulates diversity by varying factors such as the number of seed topics, prompt templates, the capabilities of the generation model, and the mixing ratio of real-to-synthetic data.

**Strengths:**

1. The challenge of evaluating and understanding the quality of synthetic data is highly relevant to the field, given the increasing reliance on such data to overcome data scarcity for LLM pre-training.
2. The paper proposes a new, LLM-based metric that attempts to move beyond simpler heuristic (e.g., n-gram) or model-based (e.g., perplexity) measures to capture a more semantic or structural form of diversity.
3. The study is well-structured, systematically isolating several key variables in synthetic data generation (topics, prompts, models, ratios). This controlled approach provides a clear framework for analyzing how these factors influence the proposed metric and, consequently, model performance.

**Weaknesses:**

1. Limited Novelty of Key Findings: Several of the paper's main conclusions, while empirically validated, align with widely accepted intuitions and existing findings. For instance, the observations that increased data diversity benefits model training, or that stronger generation models (e.g., GPT-4o vs. GPT-3.5) produce higher-quality synthetic data, are largely consistent with the prevailing understanding in the field and have been supported by previous work.

2. Lack of Explanatory Depth: The analysis of factors like prompt templates and data ratios is somewhat superficial. The paper effectively demonstrates that these factors correlate with performance but provides limited insight into why. For example, the claim in Line 431 that over-weighting synthetic data "may introduce redundancy" is a critical point that warrants a much deeper analysis. The paper misses an opportunity to investigate the nature of this redundancy (e.g., semantic, stylistic, or topical).

3. Weak Validation of the Proposed Metric: The foundational assumption of the LLM cluster-agent is that a higher number of clusters identified by the LLM (relative to sample size) directly equates to greater data diversity. However, this assumption is not rigorously validated. The primary validation experiment (Fig. 4) relies on varying the number of seed topics ($\mathcal{T}$). However, $\mathcal{T}$ measures topic variety, which is only one facet of diversity. "Diversity" is a multi-dimensional concept encompassing stylistic, structural, syntactic, and semantic variations. The paper does not sufficiently demonstrate that the LLM cluster-agent captures these other crucial dimensions. The finding that models trained on more topics perform better is, in itself, an expected outcome and does not conclusively prove the general utility of the proposed metric.

4. Overstated "Large-Scale" Claim: The claim of conducting a "large-scale study" is debatable. The experiments are limited to 350M and 1.4B parameter models. While these are non-trivial, the findings cannot be reliably extrapolated to current state-of-the-art models (e.g., 7B, 70B, or larger), which may interact with data diversity in different ways. The conclusions would be significantly strengthened by verification on at least a mid-scale model (e.g., 7B-8B).

5. Minor Issues and Presentation: The Appendix, in its current state, appears unfinished and requires significant proofreading. There are numerous large blank spaces, inconsistent formatting (e.g., between titles and boxes), and blue-colored text (e.g., Pages 20, 21) that suggest it is still a work-in-progress. This section needs to be thoroughly formatted and polished before publication. There are also some typos in the main text, e.g., line 287 should be "(h) LLM cluster score."

**Questions:**

1. Earlier research has proposed the concept "Semantic Entropy"[1], which also quantifies language diversity by clustering or classifying text data, and some further works adopt this method to evaluate the diversity of text data. Can you provide further discussion about the LLM cluster-agent and semantic entropy (such as some empirical comparison) to show the advantage of the LLM cluster-agent?

[1] Farquhar, S., Kossen, J., Kuhn, L., & Gal, Y. (2024). Detecting hallucinations in large language models using semantic entropy. Nature, 630(8017), 625-630.

2. As mentioned in the weaknesses, the claim that over-weighting synthetic data introduces "redundancy" (Line 431) is underdeveloped. Could the authors elaborate on the nature of this redundancy? Is it an artifact of the generation model (e.g., mode collapse, stylistic tics) or an inherent property of upsampling a finite (though large) set of synthetic data?

3. To strengthen the validation of the LLM cluster-agent, have the authors considered applying it to existing, benchmark datasets that are commonly accepted as having high or low diversity? For example, how does the metric score a highly curated dataset (e.g., a subset of Wikipedia) versus a noisy, uncurated web crawl? Comparing the metric's score on these known quantities would provide more convincing evidence of its validity beyond just topic coverage.

---

### Official Review · Reviewer_oAmx · 2025-10-30

**Soundness:** 2
**Presentation:** 3
**Contribution:** 2
**Rating:** 4
**Confidence:** 3

**Summary:**

This paper investigates the role of synthetic data diversity in the performance of Large Language Models (LLMs). It introduces a new LLM-based diversity metric calculated from the LLM Cluster-agent, which measures corpus-level diversity via iterative LLM clustering and self-verification. By conducting experiments on 350M and 1.4B parameter models, the study shows that higher diversity correlates positively with both pre-training and fine-tuning performance. In experiments, the paper further explores factors influencing diversity including generation prompts, model choice and the ratio of synthetic to real data. Empirical experiments provide valuable observations for effective synthetic data construction.

**Strengths:**

1. The study proposes an LLM Cluster-agent that provides a pure LLM-driven paradigm for quantifying data diversity.
2. The paper provides a collection of well-documented prompt templates. It offers practical guidance for both prompt-based clustering and data generation.
3. The experiments include a broad range of heuristic and model-based diversity baselines. It provides a comprehensive comparison that highlights the distinct behaviors of various diversity metrics.

**Weaknesses:**

1. Several findings presented in this paper have been discussed [1]. The authors should concisely highlight what new mechanisms or empirical evidence the proposed metric reveals.
2. The paper claims that using an LLM-based clustering framework to quantify data diversity is novel, but clustering-based diversity metrics are not new. The LLM cluster score may depend strongly on the scale or architecture of the LLM as well as prompt designs. Besides, the number of clusters $K$ is limited because the LLM can only cluster a small number of samples per iteration. The study does not clearly demonstrate the advantages of LLM-driven clustering.
3. In section 3.2, the ground-truth diversity of synthetic datasets would be more convincing by introducing human evaluations or additional empirical validation.
4. Experiments are restricted to small language models (350M and 1.4B) that may not fully generalize to current LLM scales. The observed trends might change for larger models or datasets.

[1] On LLMs-Driven Synthetic Data Generation, Curation, and Evaluation: A Survey. 2024.

**Questions:**

1. In Section 3.2, how does the de-duplication process ensure that topics are semantically distinct and not overly overlapping? For those semantically similar topics, GPT-4o may still generate highly similar synthetic data. Since Appendix B.2 only reports statistical results for $T=300K$, could the authors provide additional quantitative or human-evaluated evidence for different synthetic datasets?
2. As the other clustering-based metric, the k-means baseline is reported to determine the number of clusters based on a trade-off between efficiency and accuracy. In Tab. 9, it appears that synthetic datasets evaluated via the k-means metric unexpectedly degrade model performance. Could the authors provide more implementation details of the k-means baseline and analyses about the phenomenon?
3. In the ablation study, the LLM cluster score varies substantially as $K$ increases. Does this imply that the diversity measured from subsets of the synthetic data may not generalize well to the diversity of the whole dataset?
4. The paper only reports correlations between model accuracy and various diversity metrics to support the validity of the proposed metric. In Fig. 11, the accuracy may be influenced by some confounding factors like the number of tokens in each synthetic dataset. I am curious whether the correlation between the LLM’s generation temperature and the proposed diversity metric can better demonstrate the feasibility of existing metrics.

---

### Official Review · Reviewer_WWo2 · 2025-10-31

**Soundness:** 2
**Presentation:** 3
**Contribution:** 2
**Rating:** 4
**Confidence:** 4

**Summary:**

This paper investigates the critical role of synthetic data diversity in the pre-training and subsequent supervised fine-tuning of Large Language Models. The authors introduce a novel, LLM-based metric named LLM Cluster-agent, designed to evaluate the semantic and structural diversity of text corpora. Through a series of controlled experiments where diversity is manipulated via topic distribution, prompt design, and generator models, the study demonstrates a significant positive correlation between the diversity score from their proposed metric and downstream model performance. This work posits that diversity is a key predictive factor for the efficacy of synthetic data in LLM training.

**Strengths:**

1. This paper is well-written, logically consistent, and clearly structured. The authors present their arguments cogently, making the methodology and results easy to follow.
2. The study provides a thorough and detailed overview of the current landscape of synthetic data generation for LLMs, including a valuable discussion of existing diversity evaluation methods and their limitations.
3. The claims are substantiated by extensive experiments. The inclusion of detailed prompt templates and ablation studies in the appendices significantly enhances the transparency and potential reliability of the findings.

**Weaknesses:**

1. The proposed LLM Cluster-Agent metric is heavily dependent on calls to powerful LLM APIs for its core functions, such as the generation of metadata and metrics.
2. The metric appears prohibitively expensive, requiring numerous API calls for evaluation, which may make it inaccessible for many research groups and complicates reproducibility.
3. The evaluation process is inherently difficult to quantify precisely, as its effectiveness is directly coupled to the performance of the specific LLM used for the evaluation. As the paper's own ablation studies suggest the evaluation results are not consistent across different LLMs, which undermines the metric's stability and claim to being a standardized measure
4. The LLM Cluster-Agent method appears to be highly sensitive to prompt design. Its success seems to hinge more on sophisticated prompt engineering rather than a novel theoretical framework for diversity. This reliance on prompts arguably positions the work more as an  engineering application rather than a fundamental theoretical advance.
5. The experiments are primarily limited to small-scale models, namely 350M and 1.4B parameters. This is a significant deviation from the current focus on much larger foundation models. It remains unclear whether the observed positive correlations will generalize to these larger-scale models.
6. The described methodology separates the data generation and diversity measurement processes. The LLM Cluster-agent is used to evaluate a dataset after it has already been generated. This "offline" evaluation workflow is computationally costly, as it provides no feedback loop to adjust or terminate the generation process if the data is found to be of low diversity, leading to significant wasted resources.

**Questions:**

1. The paper's methodology for controlling diversity relies on proxies such as topic count, generation frequency per topic, and text style. Beyond these methods, have the authors considered other, perhaps more rigorous or fine-grained, methods for controlling the diversity of synthetic data generation?
2. The study compellingly shows that superior generator models (e.g., GPT-4o) produce data that leads to better downstream performance, even under the same diversity-control parameters as weaker models (e.g., GPT-3.5). Does the paper offer any insight into what constitutes this performance gap beyond the measured diversity score? Is it possible that the data from GPT-4o is not just more diverse, but simply of higher quality, and that the proposed metric is not fully capturing this qualitative difference? How could this "quality gap" be quantified?

---

### Official Review · Reviewer_jD9H · 2025-11-01

**Soundness:** 2
**Presentation:** 3
**Contribution:** 2
**Rating:** 4
**Confidence:** 3

**Summary:**

This paper proposes an LLM Cluster-agent metric to measure the diversity of synthetic data, and experimentally verifies its positive correlation with LLM pre-training and fine-tuning performance. The study finds that the generative model, prompts, and the ratio of real to synthetic data significantly affect diversity, with diversity having a more significant impact on fine-tuning performance, providing guidance for the use of synthetic data in LLM.

**Strengths:**

- The diversity score is reasonable: with the same number of samples, the more reasonable clusters that can be consistently formed, the higher the diversity.

- Extensive and solid experiments and analyses were conducted.

**Weaknesses:**

- In Section 3.6, the authors control the ratio between real and generated synthetic tokens by adjusting the amount of synthetic data, without keeping the total training data amount constant.

- The layout of the first row at the bottom of Figure 5 has a problem; half of it is blank and needs adjustment.

**Questions:**

- In Section 2.1, for different synthetic datasets, are the scoring metrics generated by LLM significantly different, or will they converge?

- For non-synthetic data, can the methods in Section 2.1 be used to determine its diversity?

- Does Figure 1 illustrate that scores are not significantly related to dataset size?

---

### Meta-Review · Area_Chair_yaig · 2026-01-10

**Summary:**

The paper investigates the impact of synthetic data diversity on LLM training by proposing a novel metric, "LLM Cluster-agent". all four reviewers consistently rated the paper as "4". The consensus for rejection is driven by the limited scale of the experiments, the high computational cost and instability of the proposed metric, and a perceived lack of technical novelty.

**Reviewer Concerns:**

no rebuttal were provided, so the concerns are outstanding, e.g.,


1.	Generalization: Findings on small models  may not generalize to state-of-the-art scales

2.	Metric Practicality: The metric is prohibitively expensive.

3.	Novelty & Depth: Key conclusions (e.g., diversity is beneficial) align with existing intuition, and the analysis of "redundancy" is superficial.

4.	Validation: Lack of rigorous comparison with established metrics.

**Reviewer Scores:**

it is expected that their scores would remain at 4 or move to 3 without rebuttal.

---

### Decision · Program_Chairs · 2026-01-26

Reject